# The Sea Anemone Neurotoxins Modulating Sodium Channels: An Insight at Structure and Functional Activity after Four Decades of Investigation

**DOI:** 10.3390/toxins15010008

**Published:** 2022-12-21

**Authors:** Margarita Mikhailovna Monastyrnaya, Rimma Sergeevna Kalina, Emma Pavlovna Kozlovskaya

**Affiliations:** G.B. Elyakov Pacific Institute of Bioorganic Chemistry, Far Eastern Branch of the Russian Academy of Science, 690022 Vladivostok, Russia

**Keywords:** sea anemone, neurotoxins of types 1–4, voltage-gated sodium channels, electrophysiology, molecular modeling

## Abstract

Many human cardiovascular and neurological disorders (such as ischemia, epileptic seizures, traumatic brain injury, neuropathic pain, etc.) are associated with the abnormal functional activity of voltage-gated sodium channels (VGSCs/Na_V_s). Many natural toxins, including the sea anemone toxins (called neurotoxins), are an indispensable and promising tool in pharmacological researches. They have widely been carried out over the past three decades, in particular, in establishing different Na_V_ subtypes functional properties and a specific role in various pathologies. Therefore, a large number of publications are currently dedicated to the search and study of the structure-functional relationships of new sea anemone natural neurotoxins–potential pharmacologically active compounds that specifically interact with various subtypes of voltage gated sodium channels as drug discovery targets. This review presents and summarizes some updated data on the structure-functional relationships of known sea anemone neurotoxins belonging to four structural types. The review also emphasizes the study of type 2 neurotoxins, produced by the tropical sea anemone *Heteractis crispa*, five structurally homologous and one unique double-stranded peptide that, due to the absence of a functionally significant Arg14 residue, loses toxicity but retains the ability to modulate several VGSCs subtypes.

## 1. Introduction

The search and the identification of biologically active substances produced by venomous organisms, both terrestrial and marine, determine the development and the progress of scientific fields, such as molecular and evolutionary biology, pharmacology, and medicine [1,2,3,4,5]. In recent decades, marine bioresources have been widely used to solve many problems of these fundamental and applied science. Sea anemones, one of the most ancient and preserved predatory venomous animals, are widespread at all latitudes of the ocean. 

They belong to phylum Cnidaria, class Anthozoa, subclass Hexacorallia, order Actiniaria numbering more than 1100 species [6]. They have established themselves in venomics as one of the popular sources of proteinaceous and peptide toxins. These toxins belong to different structural types and interact with a variety of biological targets. Many of them have promising pharmacological activity [6,7,8].

Sea anemones, being usually attached to the ground, lead a sedentary lifestyle and feed mainly on organic detritus, nutrients produced by symbiotic algae, zooxanthellae (single-celled dinoflagellates), green algae zoochlorellae, and by the products of their algal photosynthesis. Moreover, these marine cnidarians are also able to prey and capture crustaceans, small crabs, mollusks, sea urchins, and even small fish [9] using venom as a kind of a “remarkable cellular and biochemical weapon” [3,10], as the venomous secret (a complex cocktail of the mainly toxic components of protein and non-protein nature) causes paralysis and even death in affected preys. Therefore, sea anemones use toxins for specific purposes: protection from potential predators, attack and paralysis of preys, as well as inter- and intraspecific competition [10,11].

The production and storage of peptide toxins occur mainly in cells called cnidocytes (also known as nematocytes), equipped with unique highly specialized stinging organelles called cnidocysts (also known as nematocysts or cnidae) secreted by a Golgi apparatus and localized mostly (as previously thought) in the sea anemone tentacles [12]. However, it was later shown that the secretion of peptide toxins might occur additionally throughout the body, namely in the cells of ectodermal and endodermal glands [13]. Using transcriptome sequencing and gene ontology, it was also found that the components of the venomous secret are localized in the various organs of a polyp: tentacles, acrorhagi, actinopharynx, body column, and mesenteric filaments [5,14,15,16]. Thus, the comparison of toxin-like genes in the sea anemones *Heteractis crispa*, *Anemonia sulcata*, and *Megalactis griffithsi* showed a different expression level of various peptide components, including toxins, in tentacles, mesentery, and in the body column [15]. Toxin expression was consistently the lowest in the body column and the highest in tentacles or mesenteric fibers, depending on the sea anemone species [16]. According to some recent studies, the expression of venom components depends of the sea anemone life cycle [17]. This is provided by the morphological complexity of the venom apparatus and leads to the separation and the compartmentalization of toxic, nontoxic, or functionally distinct venom peptides in different tissues. For example, the representatives of the Nv1-like protein family of the sea anemone *Nematostella vectensis* are produced at the protein level in different life cycle stages (from eggs to primary polyps and adult females) and have various pharmacological profiles [17].

It is shown that the complex anatomical structure of cnidocysts, the number and the type of which vary in different tissues of an animal [10], is aimed at the easy penetration of venomous secret into the body of the attacked preys [14,18]. According to the results of deep sequencing of toxin transcripts belonging to five sea anemone species, toxin expression takes place in different tentacular structures. At the same time, the composition and the activity of toxins vary significantly, and the expression occurs specifically and is associated with sea anemone morphological features, as well as with the performance of certain functions in biocenosis [19,20].

A systematic study of the composition of the sea anemone venomous secrets started actively in the middle of the last century and now covers several dozen of species [1,2,3,4,5,6,7,13,15,18,21,22,23,24,25,26,27,28,29,30,31,32,33,34,35]. According to the latest data, about 236 toxic peptides have been isolated from the venoms of more than 45 sea anemone species by procedures of traditional protein chemistry (the homogenization of sea anemones, the separation of crude venom compounds and the purification of individual peptides using size exclusion, ion-exchange, and high-performance liquid chromatography). Only about 60 proteinaceous toxins have been obtained in the native state and characterized by varying degrees of depth [20]. Working in synergy, sea anemone toxic peptides have acquired an important allomonal and ecological role in marine biocenosis [8,35].

Despite an unprecedentedly huge molecular diversity of peptides produced by the Cnidaria discovered to date (belonging to 17 different molecular scaffolds), biological targets of the vast majority of sea anemone toxins remain insufficiently studied [36] (some toxins may still be unknown). Although by now it is known that the most studied representatives of the sea anemone venomous secrets are: (1) the voltage-gated sodium channel toxins called neurotoxins (NaTxs), which modulate Na_V_s [2,37,38,39,40]; (2) APETx-like peptides, inhibiting acid-sensing ion channels [41,42,43,44]; (3) voltage-gated potassium channel toxins (KTxs) blocking some K_V_ subtypes [45,46,47,48]; (4) actinoporins (or pore-forming toxins) interacting with eukaryotic cytoplasmic membrane [49,50,51]; and (5) polyfunctional protease inhibitors (mainly Kunitz-type), some of them block not only proteases, but also vanilloid receptor TRPV1, and some voltage-gated K_V_ channels [52,53,54,55,56].

The uniqueness of the sea anemones neurotoxins is mainly stipulated by their belonging to several structural types, the evolution of which has determined their high affinity and selectivity, as well as synergic action on their biological targets, different subtypes of mammals and/or arthropods/insects’ voltage-gated sodium channels (Na_V_s) possessing certain structural and physiological characteristics [7,38,40,57,58,59].

NaTxs (like those produced by scorpions [60]), were shown to disrupt cell ion conductance through the modification of Na_V_s, which are transmembrane proteins playing a significant role in the generation of action potentials in excitable and non excitable cells (cardiac and skeletal muscle cells as well as sensory neurons) [61,62,63,64]. Due to the ability of modulating Na_V_s functional activity, sea anemone neurotoxins are observed to help invaluably in the study of these channel subtypes involved in the physiological and pathological functions of cardiovascular and neurological systems.

This review considers some results of the structure-functional investigations of known to date sea anemone neurotoxins modulating voltage-gated sodium channels (NaTxs), including natural toxins of *Heteractis crispa*, as well as demonstrating some comparative aspects of their structure, toxicity, and the molecular mechanism of an interaction with different Na_V_ subtypes, evaluated by electrophysiology and molecular modeling methods.

## 2. Voltage-Gated Sodium Channels as Main Targets of Sea Anemone Neurotoxins 

Back in the 1980s [21,23,24,25,26,27,28,29,30], it was established that voltage-gated sodium channels are biological targets of sea anemone neurotoxins. Eukaryotic Navs are complexly organized transmembrane single-chain polypeptides, which include the highly conserved pore-forming α-subunit (220–260 kDa, 1800–2000 amino acid residues (aa)) associated with auxiliary regulatory β-subunits (β1–β4, 30–40 kDa, ~200 aa) [63]. The α-pore-forming subunit forms four highly homologous repeating structures, domains I–IV, through which the selective diffusion of ions occurs, providing the electrical activity of cells and playing a special and decisive role in excitability and neuromuscular transmission [65,66,67]. The β-subunit, according to the electron microscopy (EM) data [68], facilitates a channel localization in the cell membrane and also modulates Na_V_ properties.

Each domain of the α-pore-forming subunit contains four transmembrane helical segments (S1–S4) forming a voltage sensor, and the two other segments (S5 and S6) participating in the formation of an ion-conducting pore (pore domain PD), permeable to sodium ions [37,38,59,61]. Voltage-gated sodium channels are responsible for the initiation and propagation of action potentials in excitable cardiac, nerve, and skeletal muscle cells [68]. The membrane potential regulates the activation of Na_V_s, and polarization triggers conformational changes causing the rapid entry of Na^+^ into the cell through channel pores.

Currently, there are nine different α-subunits of human Na_V_s (Na_V_1.1–Na_V_1.9 [59,69]) encoded by the gene family (SCN1A–SCN5A and SCN8A–SCN11A). Moreover, each subtype can have several isoforms determining its localization, characteristics of currents passing through Na_V_ pores, and participation in certain physiological and pathological processes [38,40,57,63,70,71]. The channel isoforms Na_V_1.1, 1.2, 1.3, and 1.6 have been observed to be expressed primarily in the central nervous system (CNS); while Na_V_1.7, 1.8 and 1.9 are expressed in the peripheral nervous system (PNS), and Na_V_1.4 and 1.5 isoforms are found in skeletal and cardiac muscles, respectively. It has been established that mutations in certain sodium channel isoforms lead to changes in electrophysiological properties, resulting in negative effects on the functional activity of Na_V_, dysfunctions and critical disturbances of the nervous, cardiovascular, and musculoskeletal systems, and in the case of such human diseases as periodic paralysis, epilepsy, chronic pain syndrome, long QT syndrome, and arrhythmia, etc. [72,73,74,75,76].

It is found that sea anemone NaTxs modulate the functional activity of Na_V_s by interacting with its α-subunit. Binding to the extracellular so-called ‘receptor site 3’ (a loop between segments S3 and S4 in the voltage-sensing domain IV (VSD-IV)) (Figure 1), all known NaTxs block segment S4 in its internal position. This results in the inhibition of Na_V_ conformational changes necessary for a fast inactivation and prolongs the open channel state during the depolarization procedure, which is associated with Na_V_ inactivation (the transition from an open to the closed state) [24,64,73,74,75,76,77,78]. Thus, S4 segment plays an important and decisive role in the fast Na_V_ activation and inactivation [77], and also performs a specific role in the voltage-dependent regulation and functional activity of sodium channels [72]. This ligand-channel interaction slows down the inactivation kinetics and prolongs the action potential of excitable and non-excitable cell membranes in a potential-dependent manner, causing the significant release of neurotransmitters at synapses [24,57,59,62,64,77,78,79]. The permeability of Na_V_ for Na^+^ ions and low molecular weight modulators is also regulated by changes in the conformation of amino acid residues of S6 segment, which form the intracellular activation gate (or a-gate) mechanism [78]. It has been found that NaTxs do not participate in the interaction with Na_V_ β-subunit co-expressed with α-subunit. Thus, by controlling the opening and closing processes of the various sodium channel subtypes, NaTxs regulate electrical signals that transmit vital information throughout a living organism [61,67].

The process of slowing down the kinetics of Na_V_ currents inactivation has been studied over the past few decades by a number of authors [27,40,80,81,82,83,84,85]. It is not surprising that NaTxs have been applied for studying Na_V_ sensing mechanism and function [2]. Nav channels as the basis for the functioning of the nervous system have been the subject of an unrelenting interest of researchers over the past decades. In 1963, A.L. Hodgkin and A.F. Huxley received the Nobel Prize in Physiology or Medicine for the development of a mathematical model describing the emergence and propagation of action potentials, and the role of VGSCs in their formation. These discoveries were the basis for the scientific achievements of the Nobel Prize laureates E. Neer, B. Sackmann (1991) and R. McKinnon (2003), who studied the structure and the function of single ion channels in cells (by the patch clamp method), as well as the operation of ion channels [86].

Significant efforts of researchers in recent decades have been aimed at studying the mechanisms that control the functional activity of sodium channels in sensory neurons, cardiac, and skeletal muscle cells. Sea anemone neurotoxins, which play an important role in the physiological or pathological functions of the Na_V_s, provide invaluable assistance in obtaining information about the mechanisms of functioning of the CNS.

## 3. Structure-Functional Characteristics of Sea Anemone NaTxs

### 3.1. Structural Types of NaTxs

Today, it is quite clear that the wide range of modulating action of NaTxs and the effect on the action potential duration of Na_V_s are due to their structural characteristics, surface electrostatic potential, and dipole moment [81,87]. Sodium channel toxins produced by sea anemones, neurotoxins, are currently the best characterized [1,2,4,7,13,14,22,35,37,39,58,80,88,89]. Traditionally all NaTxs studied to date have been classified into four structural types, according to amino acid sequences and a different Cys residues distribution pattern (Figure 2, Figure 3, Figure 4 and Figure 5) [2,7,80,81,90]. The experimental data have demonstrated that the toxic and paralyzing action of sea anemone venoms is associated with the profound effects of neurotoxins type 1 and 2 (though with different toxic activity in the concentration range from a few ng to hundreds of μg) on prey, with which animals communicate in natural conditions [2,3,21,22,23,24,25,26,27,28,29,30,31,32,33,34,80,81,82,91,92,93,94].

Almost all type 1 and 2 natural NaTxs produced by sea anemones possess from 46 to 49 amino acid residues linked by three disulphide bridges, following the similar connectivity pattern (C1-C5, C2-C4, C3-C6) not yet found in any other toxin (Figure 2 and Figure 3). The only exception is type 1 NaTxs isolated from the sea anemone *Actinia equina*, Ae1, which has 54 amino acid residues [33]. In addition to Ae1, a number of its highly identical amino acid sequences (Ae2-1, Ae3-1, Ae4-1, Ae2-2, Ae2-3) have been derived from *A. equina* mRNA sequences [95].

Today, more than 40 type 1 NaTxs are known (Figure 2); most of them have been isolated from the species belonging to the Actiniidae family (Anthopleura, Anemonia, Condylactis, Bunodosoma, Actinia, Antheopsis genera) [2,81]. Besides several *A. equina* neurotoxins, some amino acid sequences of putative homologs of the native neurotoxin Av2 have been derived from the sea anemone *A. viridis* gDNA sequence [88,95]. Gene cloning, which became a traditional method of studying sea anemone neurotoxins by the end of the last century, made it possible to detect up to a dozen or even more highly homologous amino acid sequences in each of the studied species, which significantly expanded the arsenal of neurotoxins for further structural and functional studies [5,14,16,39]. The majority of the studied sea anemone neurotoxins to date belong to the most common structural type 1: ATX-Ia, -II, -V from *Anemonia sulcata* [21,22,23,24,26,82,93,94,95,96,97,98], ApA and ApB from *Anthopleura xanthogrammica* [25,30,99], Am-III from *Antheopsis maculata* [100], Rc-1 from *Radianthus crispus* [101], CgNa from *Condylactis gigantea* [102], Gigantoxin-2 from *Stichodactyla gigantea* [103], BcIII from *Bunodosoma caissarum* [104], BgII, BgIII from *Bunodosoma granulifera* [105], and cangitoxins from *Bunodosoma cangicum* [106,107,108].

**Figure 2 toxins-15-00008-f002:**
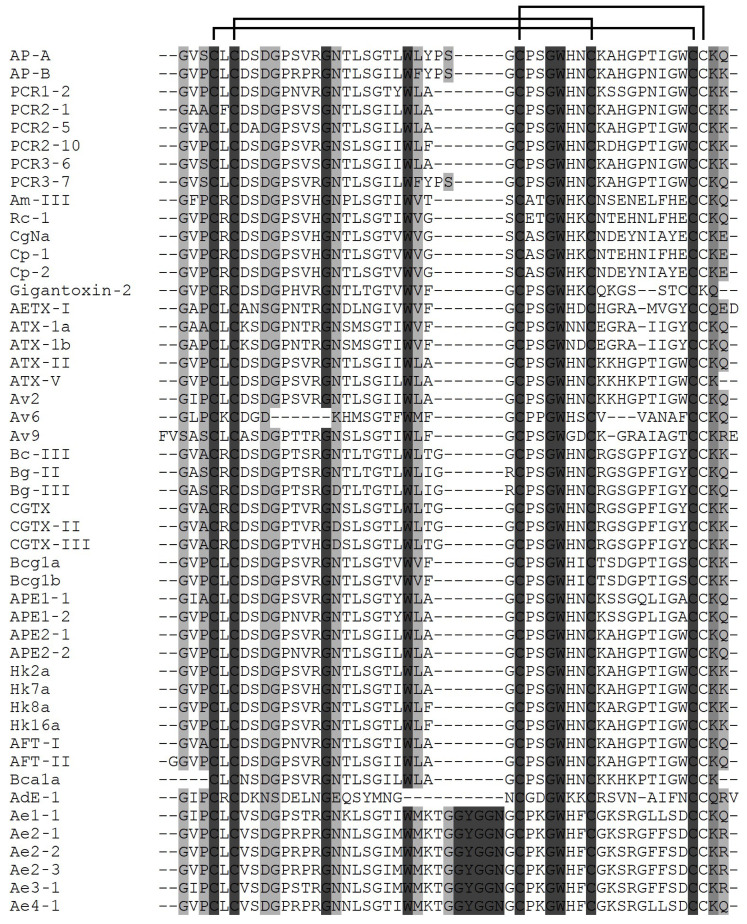
Multiple alignment of the amino acid sequences of the type 1 sea anemone NaTxs: ApA (UniProt ID: P01530) [25], ApB (P01531) [30], PCR1-2 (P0C5F8), PCR2-1 (P0C5G0), PCR2-5 (P0C5F9), PCR2-10 (P0C5G1), PCR3-6 (P0C5G2), PCR3-7 (P0C5G3) [99] from *A. xanthogrammica*; Am-III (P69928) [100] from *A. maculata*; Rc-1 (P0C5G5) [101] from *Heteractis crispa* (=*R. crispus*); CgNa (P0C280) [85,102] from *C. gigantea*; Cp1 (P0CH42) and Cp2 (P0C280) [32] from *Condylactis passiflora*; Gigantoxin-2 (Q76CA3) [103] from *S. gigantea*; AETX-1 (P69943) [34] from *Anemonia erythraea*; ATX-Ia (=ATX-I) (P01533) [96], ATX-Ib (A0A0S1M165), ATX-II (P01528) [23], ATX-V (P01529) [97] from *A. sulcata*; Av2 (P0DL52) [95,98] from *Anemonia viridis* (previously known as *A. sulcata*) and Av6, Av9 (sequences, deduced from *A. viridis* genomic DNA [95]; BcIII (Q7M425) [104] from *B. caissarum*; BgII (P0C1F4), BgIII (P0C1F5) [105] from *B. granulifera*; CGTX (P82803), CGTX-II (P0C7P9), CGTX-III (P0C7Q0) [106], Bcg1a (P86459), Bcg1b (P86460) [107,108] from *B. cangicum;* APE1-1 (P0C1F0), APE1-2 (P0C1F1) [109], APE2-1 (=ApC) (P01532) [110], APE2-2 (P0C1F3) [109] from *Anthopleura elegantissima*; Hk2a (P0C5F4), Hk7a (P0C5F5), Hk8a (P0C5F6), Hk16a (P0C5F7) [111] from *Anthopleura* sp.; AFT-I (P10453), AFT-II (P10454) from *Anthopleura fuscoviridis* [112]; Bca1a (GenBank accession number: KY789430) [113] from *Bunodosoma capense*; AdE-1 (E3P6S4) [114] from *Aiptasia diaphana*; Ae1 (=Ae1-1) (Q9NJQ2) [33] from *A. equina*, Ae2-1 (B1NWU2), Ae2-2 (B1NWU3), Ae2-3 (B1NWU4), Ae3-1 (B1NWU5), and Ae4-1 (B1NWU6) derived from *A. equina* genomic DNA [95]. The disulfide bridges are shown above the alignment. The sequence similarity is shown as a dark (high) and light (low) gray background, the multiple sequence alignment was performed using the Vector NTI Advance 11.0 software.

In addition to these neurotoxins, 14 NaTxs type 2 representatives from several species of the family Stichodactylidae (mainly Stichodactyla and Radianthus (=Heteractis) genera) have also been isolated to date [27,28,29,115,116,117,118,119,120]. Recently, three new representatives of type 2 neurotoxins, δ-TLTX-Hh1x (Hh X), δ-TLTX-Ca1a (Ca I), and δ-TLTX-Ta1a (Ta I), toxic to crabs, have been discovered in sea anemones belonging to the Thalassianthidae family species (*Heterodactyla hemprichii*, *Cryptodendrum adhaesivum*, and *Thalassianthus aster*) (Figure 3) [119]. The complete amino acid sequences of type 2 neurotoxins have been established using protein structural chemistry (neurotoxin sequencing by Edman degradation) and molecular biology (cloning and sequencing of genes coding some neurotoxins).

**Figure 3 toxins-15-00008-f003:**
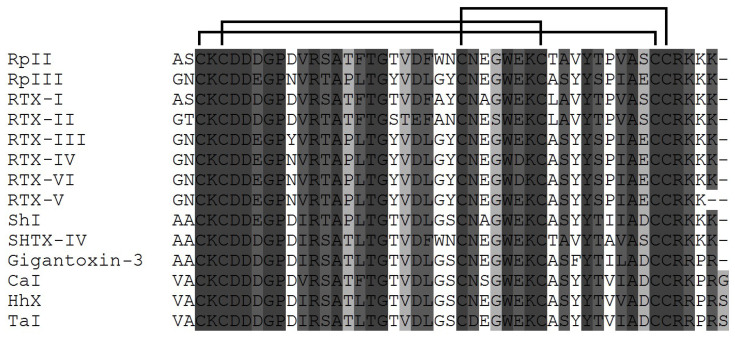
Multiple alignment of the amino acid sequences of type 2 sea anemone NaTxs: RpII (UniProt ID: (P01534) [27], RpIII (P08380) [91] from *Heteractis magnifica* (*Radianthus paumotensis*); RTX-I (P30831) [28], RTX-II (P30783) [115], RTX-III (P30832) [29], RTX-IV (P30784), and RTX-V (P30785) [116] from *Heteractis crispa* (earlier *Radianthus macrodactylus)*; SHTX-IV (B1B5I9) [117] from *Stichodactyla haddoni*; ShI (P19651) [31,118] from *Stichodactyla helianthus*; Gigantoxin-3 (Q76CA0) [103] from *S. gigantea*; Ca I (D2KX90) from *C. adhaesivum*, Hh X (D2KX91) from *H. hemprichii*, Ta I (D2KX92) [119] from *T. aster.* The disulfide bridges are shown above the alignment. The sequence similarity is shown as a dark (high) and light (low) gray background, the multiple sequence alignment was performed using the Vector NTI software.

The NaTxs belonging to types 1 and 2 are believed to have the same ancestral gene [2]. The representatives of these types have proved to differ immunologically as antigenic cross-reactivity has not been revealed for them [27,31]. It has been found that both NaTx types, 1 and 2, have the same binding site (so-called ’receptor site 3‘) [120], the extracellular region of Na_V_ voltage-sensor domain IV (VSD-IV) [77,121]. It is one from the eight known Na_V_ binding sites. It is also the binding site of scorpion α-toxins [24]. However, the interaction of type 2 NaTxs with the site 3 (Figure 1), in contrast to that of type 1 NaTxs, does not block the binding of α-toxins to this site [110]. Moran et al. (2007) believe that this site is a heterogeneous receptor for many so-called site-3 toxins, including sea anemone neurotoxins [122,123].

Unlike types 1 and 2 neurotoxins, five shorter NaTxs (with 27–32 amino acid residues and toxicity for crabs) were mainly isolated from the sea anemones of the Actiniidae family (Figure 4**a**) and assigned to type 3 [110]. Thus, the shortest neurotoxin ATX-III (=Av3, 27 aa) isolated earlier from *A. sulcata* (=*A. viridis*) [70,124,125], is stabilized by three disulfide bonds (C1-C5, C2-C4, C3-C6). While four others longer neurotoxins of this type, PaTX (31 aa) from *Entacmaea quadricolor* (*=Parasicyonis actinostoloides*) [126], Da-I and Da-II (30 aa) from *Dofleinia armata*, and Er-I (32 aa) from *Entacmaea ramsayi* [127], were observed to contain four disulfide bonds. These five toxic peptides are unrelated to each other and evidently have different structural folds and properties [57,122]. According to ^1^H-NMR data and the secondary structure elements analysis, ATX-III forms an unusual compact structural motif with the complete absence of regular α-helices and β-strands, but only with β- and γ-turns [128]. The residues localized on the molecular surface form a hydrophobic region and may represent a Na_V_ binding surface [129]. Like NaTx types 1 and 2, short type 3 neurotoxins also slow down the kinetics of Na_V_ channel inactivation [82,110]. They were shown to be inactive [21,26] on mammalian cells and very active on that of crustaceans and insects [65,70,122,130,131]. The recombinant form of neurotoxin Av3 (=ATX-III) was also inactive on mammalian rNa_V_1.2a, rNa_V_1.4, and hNa_V_1.5 channels [122]. It is still unknown how other type 3 neurotoxins interact with insect and mammalian Navs.

The small sea anemone *Calliactis parasitica* (Hormathiidae family), inhabiting marine biocenosis in symbiosis with hermit crabs, produces two highly identical toxins, calitoxins I and II (CLX-1 and CLX-2) (46 aa, 95% identity), classified as type 4 NaTxs (Figure 4b) [132,133]. Their residues are cross-linked by three disulphide bridges, the topology of which is similar to that of types 1 and 2 NaTxs but, despite this, they are attributed to the separate type 4 due to differences in amino acid sequences with those of types 1 and 2 representatives. Both of the peptides differ in the substitution of only one amino acid residue at the position 6 (Glu/Lys). Neurotoxins of types 3 and 4, whose amino acid sequences differ significantly from those of types 1 and 2, appear to be less abundant in sea anemone venoms [31,57,134].

In addition to these four types, two native neurotoxins are described: Halcurin isolated from the sea anemone *Halcurias carlgreni* (a primitive specie belonging to the Halicuridae family [135]) and Nv1 from *Nematostella vectensis* (from the small starlet sea anemone of the Edwardsiidae family) showing higher specificity for insect channels [136]. The amino acid sequences of both neurotoxins (47 aa) are highly homologous to those of NaTxs types 1 and 2 (Figure 4c).

**Figure 4 toxins-15-00008-f004:**
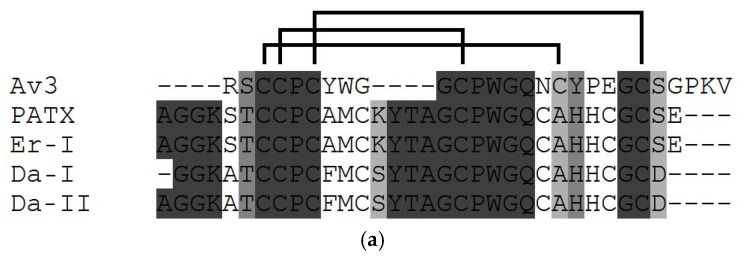
Multiple alignment of the amino acid sequences of: (**a**) type 3 sea anemone NaTxs: ATX-III (=Av3) (UniProt ID: P01535) from *A. sulcata* [124], PaTX (P09949) from *E. quadricolor* (=*P. actinostoloides*) [126], Er-I (P09949/P09949-1) from *E. ramsayi*, Da-I (a major form) and Da-II (P0DMZ2/P0DMZ2-1) from *D. armata* [127]; (**b**) type 4 NaTxs CLX-1 (UniProt ID: P14531) and CLX-2 (P49127) from *C. parasitica* [132]; and (**c**) Halcurin (UniProtKB: P0C5G6) from *H. carlgreni* [135] and Nv1 (B1NWS1) from *N. vectensis* [137]. The sequences of two Nv1 isoforms with C-terminal Q (B1NWS1-1) and K (B1NWS1-2) are shown [https://www.uniprot.org/uniprot/B1NWS1] [138]. C-terminal residues of both isoforms are underlined. The disulfide bridges are shown above the alignment. The sequence similarity is shown as a dark (high) and light (low) gray background, the multiple sequence alignment was performed using the Vector NTI software.

The high activity of Nv1 has been established on arthropods Na_V_s in a way that this neurotoxin inhibits the inactivation of DmNa_V_1/TipE channel of drosophila by binding to its site 3. At the same time, Nv1 is almost inactive on mammalian channels (rNa_V_1.4-β-1, hNa_V_1.5-β-1). It is completely inactive on the rNav1.2a-β-1 channel [137]. The dozen of putative Nv1 neurotoxins derived from the cluster of genomic DNA was presented by Moran et al. [95]. Furthermore, some genes encoded identical precursors of this neurotoxin. Moreover, some more peptides belonging to the Nv1 family (Nv4–Nv8) encoded by highly homologous genes at different *N. vectensis* developmental stages were obtained later [17]. The toxicity of different representatives of this family varied for different organisms, zebrafish larvae, and cherry shrimps [17]. The residues of these neurotoxins are cross-linked by three disulphide bridges similarly to the bridges of the type 1 and 2 representatives (C1-C5, C2-C4, C3-C6). A high degree of Halcurin sequence homology with the type 2 neurotoxins (49–74%) is revealed. This neurotoxin has several residues conserved only for type 1 NaTxs, and also has 47% sequence identity with Nv1. Halcurin has been observed to be very toxic to crabs (with LD_50_ of 5.8 μg/kg), but it has not been lethal to mice.

Besides these sea anemone representatives, there are neurotoxins with lethality to crabs, AETX-II and AETX-III (Figure 5), isolated from the small sea anemone *Anemonia erythraea* (Actiniidae family) (an organism with a body wall diameter of 2–5 cm) inhabiting tropical ocean regions [34]. Their amino acid sequences include 59 residues (with more than 90% identity) and a distinct fold is stabilized by five disulfide bonds located at the positions 5, 11, 12, 19, 27, 28, 36, 44, 46, 60, a disulfide pattern of which is not determined yet. Although the mechanism of modulating action of these neurotoxins has not yet been studied, the sequence homology (40%) with the toxin Tx1 from the *Phoneutria nigriventer* spider indicates the likelihood of the similar modulating effect of AETX-II and AETX-III towards voltage-gating Na_V_s. It has been established that neurotoxins are inactive on mammalian channels. Moreover, the LD_50_ values of AETX-II and AETX-III against crabs have been estimated to be 0.53 and 0.28 µg/kg, respectively [34].

**Figure 5 toxins-15-00008-f005:**
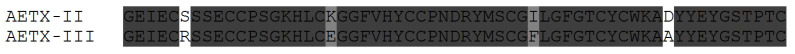
Amino acid sequences of AETX-II, AETX-III from *A. erythraea* [34]. The sequence similarity is shown as a dark (high) and light (low) gray background.

Three new toxins, AnmTX Cj 1a-1 (48 aa), AnmTX Cj 1b-1 (46 aa), and AnmTX Cj 1c-1 (49 aa), have recently been identified from the cold-water sea anemone *Cnidopus japonicus* by high-throughput proteomics and bioinformatics methods [139]. Both toxins, AnmTX Cj 1c-1 and AnmTX Cj 1a-1, are homologous to NaTxs types 2 and 1, gigantoxin-2 (44 aa), and gigantoxin-3 (49 aa) from *S. gigantea* with 52 and 50% homology, respectively (Figure 6a). Apparently, the pattern of their disulfide bonds is similar to that of type 1 and 2 neurotoxins. Moreover, the AnmTX Cj 1b-1 toxin is homologous to BDS-I (43 aa) from the sea anemone *A. sulcata* [140] (Figure 6a) and their sequences homology with each other is 48%.

To date, the specific targets of gigantoxin-3 and gigantoxin-2, as well as AnmTX Cj 1a-1 and AnmTX Cj 1c-1, have not yet been identified. However, AnmTX Cj 1a-1 and AnmTX Cj 1c-1 homology to some types 1 and 2 neurotoxins supports their functional activity as modulators of sodium channels inactivation. It has recently been found that BDS-I, known as a specific inhibitor of the K_V_3 family of voltage-gated potassium channels [48] (Figure 6), is also able to bind to human Na_V_1.7 channels [140] with very high binding efficiency (similar to that of the neurotoxin 1 type CGTX-II from *B. cangicum* (Figure 6b) [106]).

In addition to the above sea anemone neurotoxins, two new toxic peptides, BcIV (41 aa) [141] and Am-II (46 aa) [100], were found in the venoms of the sea anemones *B. caissarum* and *A. maculata*, respectively. They turned out to be homologous to the peptide toxins APETx1 (42 aa) [41] and APETx2 (42 aa) [42] from *A. elegantissima*, as well as BDS-I (43 aa) [140], and BDS-II (43 aa) [142] toxins from *A. sulcata* (Figure 6c).

Both peptides, BcIV and Am-II, are characterized by a β-defensin-like fold similar to that of NaTx neurotoxins and APETx-like peptides, which have the same pattern of disulfide bonds. However, the identity of the amino acid sequences of BcIV and Am-II with the sequences of known NaTxs (Figure 2) and APETx-like (Figure 6c) representatives is somewhat lower (BcIV has 45 and 48% identity with APETx1 and APETx2 as well as 42% with Am-II and BDS-I, BDS-II (Figure 6a–c)).

It has been shown that BcIV is able to bind Na_V_ channels because, like the type 1 neurotoxin BcIII [104], it has a paralyzing effect in vivo [141], albeit in a much lower minimal lethal dose comparing to BcIII (2000 μg/kg vs 219 µg/kg, respectively); while the main targets of APETx1 and APETx2 peptides, as known, are the human acid-sensing ion ASIC3 channel and human cardiac potassium channel, hERG [41,42]. It is known that APETx2 effectively inhibits, besides ASIC3, Na_V_1.2 (IC_50_ of 114 ± 25 nM) and Na_V_1.8 channels (IC_50_ of 55 ± 10 nM).

The new Am-II peptide also has a paralytic effect (LD_50_ 420 µg/kg) six times lower than its homolog from *A. maculata*, Am-III (LD_50_ of 70 µg/kg), the representative of NaTxs type 1, and twice higher than Am-I (LD_50_ of 830 µg/kg) structurally similar to PaTX-like NaTxs type 3 (Figure 4a) [100]. However, unlike BDS-I and BDS-II, which highly specifically blocks the activity of the potassium channel K_V_3.4 and has a weak effect on TTX-sensitive Na_V_ of neuroblastoma and cardiac and skeletal muscle cells, Am-II, very likely, acts on Na_V_ channels as it is lethal to crabs [100]. Therefore, BcIV and Am-II are structurally new peptide toxins with a putative species-specific action.

### 3.2. Secondary and Spatial Structures of NaTxs

The search and study of NaTx biological targets, and the elucidation of their structure-functional relationships, are impossible without the determination of the spatial structure of neurotoxins of all structural groups. The 3D structure, in turn, is the basis for studying the mechanisms of protein-ligand interactions. The spatial structures of type 1 neurotoxins such as ApA [143], ApB [144], ATX-I [145], and CgNa [102], as well as the type 2 neurotoxin ShI [118,146], and the type 3 toxin, ATX-III [128,129], were established by NMR spectroscopy. The first two mentioned types adopted a unique β-defensin-like fold found only in sea anemone NaTxs of β-type, which is a twisted antiparallel four-stranded (in most representatives) β-sheet stabilized by three disulfide bonds and three well-defined loops [118]. One of the loops, elongated and conformationally highly flexible, contains conserved arginine (so-called ‘arginine loop’ or ‘Arg14 loop’ consisting of 7–18 aa), and connects β-strands 1 and 2 (Figure 7a–f,h,i). It is shown that Arg14, being the most conserved neurotoxin residue, is functionally significant for binding to sodium cannels [147]. Despite some differences in amino acid sequences, the identical localization of S-S bonds and the presence of conservative residues in both types of neurotoxins permit to that they have a common ancestral gene [2,135].

Although the 3D structure of the majority of NaTxs representing types 1, 2, and 4 has not yet been determined experimentally, high amino acid sequences identity and a common defensin-like fold stipulate their high conservation and structural similarity [148]. Despite the high similarity of neurotoxins amino acid sequences and 3D structures, their binding mechanisms to different sodium channel types may vary greatly [87]. Data on the secondary structure elements (alpha-helices and beta-sheets) of type 3 short neurotoxins are not known. According to [105], these peptides form compact spatial structures consisting of several turns (Figure 7g).

**Figure 7 toxins-15-00008-f007:**
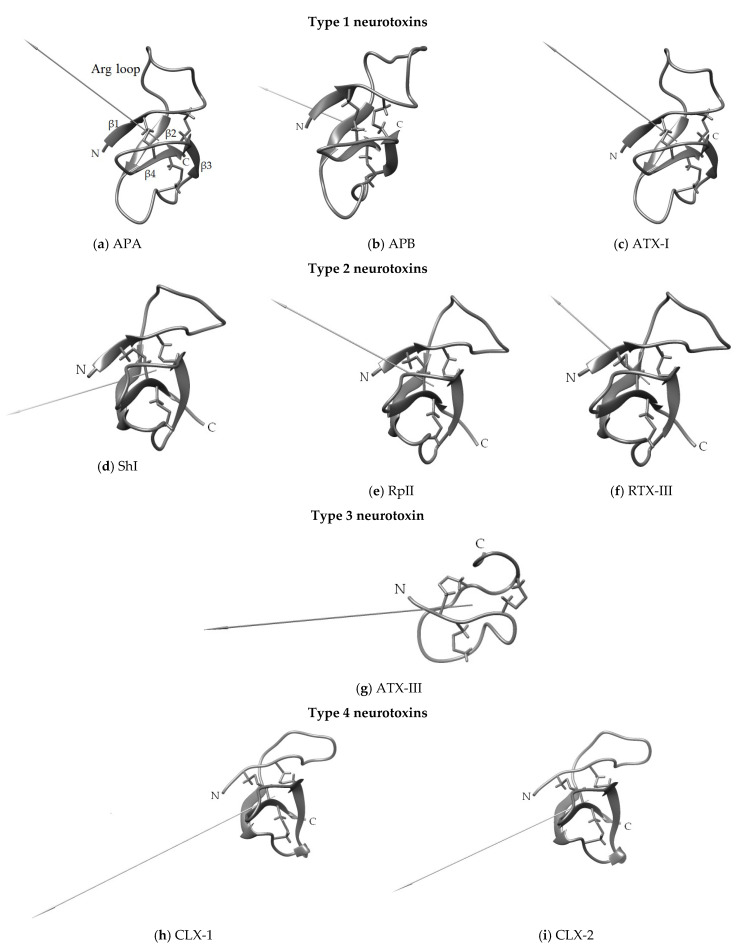
The ribbon diagrams of the 3D models of structural types 1–4 neurotoxins adopting a β-defensin-like fold. Disulfide bonds are shown by sticks. The spatial structures of the neurotoxins (ApA (PDB ID: 1Ahl) (**a**), ApB (1Apf) (**b**), ATX-I (1Atx) (**c**) (prototype ApA PDB ID: 1Ahl), ShI (2ShI) (**d**), RpII (**e**), RTX-III (**f**), ATX-III (1Ans) (**g**), CLX-1 (**h**), and CLX-2 (**i**)) are created by their homology modeling with ShI spatial structure obtained from its NMR one. Dipole moments are shown as arrows; the magnitude of dipole moments is indicated in Debye (D): 133 (**a**), 168 (**b**), 124 (**c**), 109 (**d**), 106 (**e**), 143 (**g**), 183 (**f**), 227 (**h**), and 179 (**i**). Visualization is performed with the UCSF Chimera 1.11.2rc software [149].

As shown in Figure 7, the dipole moments of neurotoxin molecules differ in both magnitude and direction. This probably explains differences in the activity of neurotoxins and their ability to interact with different subtypes of Na_V_ channels. It is quite obvious that the differences between the molecular surfaces of neurotoxins are stipulated by the presence of different functionally significant amino acid residues ensuring the preservation of the neurotoxins activity, in relation to different channel subtypes [37,80,122].

### 3.3. Some Key Functionally Significant Amino Acid Residues of NaTxs

In the 1960–1980s, it was found that sea anemone neurotoxins (such as ApA, ApB, ATX-II, ATX-III) were toxic to arthropods/insects and could also have cardiostimulatory and neuromuscular effects on mammals [21,22,23,24,25,26,30], whereas their specific effect on Na_V_ channels was later observed to manifest itself in a wide range of concentrations [34,65,66,82,83,85,93,94,103,123,150,151].

Although a large number of sea anemone neurotoxins are now known, only a few of them have been studied in sufficient detail. High sequence identity and the presence of conservative, functionally important amino acid residues, indicate the preservation of the high stability of a NaTxs structure and function [152]. The analysis of the structure-functional relationships of some NaTxs type 1 representatives, and their functionally significant and conserved amino acid residues identified by site-directed mutagenesis, has been demonstrated in a number of experimental papers and reviews [2,37,80,81].

While most type 1 neurotoxins contain 47 amino acid residues (except ATX-I and CgNa with 46 aa ApA and ApB with 49 aa, AFT-I, AFT-II and Gigantoxin with 48 aa) all type 2 neurotoxins described to date have 48 amino acid residues (Figure 2 and Figure 3). Within each of the four types of neurotoxins (1–4), there is a high (up to 98%) degree of structural homology.

A characteristic of the type 2 neurotoxins is the presence of only two mainly hydrophobic residues at the N-terminus of a sequence and of three-four basic residues at the C-terminus, which distinguishes them from type 1 neurotoxins with three N-terminal and two C-terminal residues (with the exception of ATX-V with one C-terminal Lys (Figure 2)). In addition, the amino acid sequences of both types contain different amounts of charged and aromatic residues. Therefore, the type 2 neurotoxins have seven positively and negatively charged residues (excluding RTX-I and RpII with six of them) in a sequence, and their N-terminal fragments contain much more negatively charged residues (including the conservative tripeptide 6DDD/E8), and C-terminal fragments include more positively charged ones. At the same time, almost all representatives of these NaTxs are characterized by the presence of a unique C-terminal 4-membered fragment, 45RKKK48. Therefore, the positive charge of type 2 neurotoxins is significantly higher than that of type 1.

It should also be noted that there are significant differences (playing, as known, an important role in protein-protein interactions) in the content of NaTx aromatic residues. Both neurotoxin types have one highly conserved Trp at position 28; it is unique for type 2 neurotoxins (an exception is RpII with one more Trp at position 24). While type 1 neurotoxins either have only one Phe and/or Tyr residue and two or three Trp residues, type 2 neurotoxins have two Phe and from one to five Tyr residues. This, together with a large number of charged residues at N-terminus (4, 6–8 positions) and C-terminus (45–48 positions), may cause their greater affinity to their targets, Na_V_ channels of various subtypes (due to the emergence of several intermolecular π-π interactions between them).

Currently, many type 1 neurotoxins (for example, ATX-II, ATF-II, Bc-III, Cp1, CgNa, CGTX-II, Bcg1a, and Bcg1b) have electrophysiologically been characterized in terms of their modulating activity and selectivity towards the mammals and insects sodium channels of various subtypes, Na_V_1.1–Na_V_1.9 [59,62,77,81,82,83,84,90,92,94,102,105,110,120,123,150,153,154,155,156]. So far, not more than two or three NaTxs of structure type 2 have been investigated, using this approach. It is quite obvious that point substitutions in types 1 and 2 neurotoxin sequences and, in particular, the different distribution of positively and negatively charged residues can significantly change the value of the peptides dipole moment, which results in their specific interaction with certain targets, various Na_V_ subtypes [81,87].

The first isolated neurotoxins (ApA, ApB, ATX-II, ATX-III, RTX-III, RpII [21,22,23,24,25,26,27,28,29,30,31,32,33,34]) showed high toxicity to mammals and/or arthropods (Table 1). Their further electrophysiological studies indicated a high activity in relation to various Na_V_ subtypes when these neurotoxins slowed down process of the channel inactivation. ApA was selective for the mammalian Na_V_1.5 channel, and ApB had high affinity for the variety of Na_V_ channel subtypes [147,157]. Those neurotoxins differed in seven amino acid residues and showed a 20–50-fold preference for cardiac channels over neuronal ones [158,159]. Several other neurotoxins were powerful insectotoxins (type 1 ATX-I [21,22] and type 3 ATX-III (=Av3) from *A. sulcata* (=*A*. *viridis*) [122]) while some NaTxs acted on the channels of both mammals and arthropods (for example, ATX-II from *A. sulcata* [110], RTX-I–RTX-V from *R. macrodactylus* [28,29,115,116]). Moreover, while ATX-II was characterized by a high insecticidal activity, the toxicity of highly homologous RTX-I–RTX-V (Figure 3) was higher for mammals, but it differed by 2–3 orders of magnitude [28,29,115,116,160,161]. A comparative study of the toxicity of types 1 and 2 NaTxs, CgNa (CgII) (*C. gigantea*), Cp1 (*C. parasitica*), and ShI (*S. helianthus*), respectively, showed that they were lethal to crustaceans, moderately toxic to insects, and non-toxic to mammals [85]. Since lots of sea anemones neurotoxins are highly toxic to insect channels, they can be considered a promising source of new insecticides, and the leading position in this direction belongs to the recombinant neurotoxins Av1 (*A. viridis*) and Nv1 (*N. vectensis*) [37,95]. A distinctive feature of the modulating effect of sea anemone neurotoxins on insect Na_V_ channels is the almost complete absence of channel inactivation after the toxin application, which is not typical for mammalian channels [37,80,150].

It is interesting that the results of mutagenesis and electrophysiological testing of a number of NaTxs show that the amino acid residues that are functionally important for the binding of neurotoxins to various mammalian and arthropod Na_V_s sometimes differ. For different representatives they often contradict each other. Thus, while the positively charged residues of ATX-II, Lys36, Arg14, and the negatively charged ones, Asp7, Asp9, and C-terminal Gln47 (Figure 8a), have been observed to play an important role for the functional activity as their substitutions, resulting in the abolishing of the neurotoxins effect [162]; while according to Moran et co-workers (2006), the replacement of Asp7Ala is not critical for the Av2 (=ATX-II) activity [163]. However, while Lys14, a conservative residue almost in all type 1 neurotoxins, was not essential for the ApA activity, it and positively charged Lys37 and Lys49 (Figure 8b) were observed to be functionally important for the activity or the affinity of ApB [164,165,166,167] with some subtypes of Na_V_ heart and neurons.

**Table 1 toxins-15-00008-t001:** Toxicity of NaTxs and binding to synaptosomes.

See Anemone Species	Neurotoxin	Lethal Dose, (μg/kg) ^a^		Binding to Synaptosomes of Rat Brain, K_D_ (μM) ^b^	Ref.
		LD_50_ (Mice)	LD_100_ (Crabs)		
*A. sulcata*	ATX-I ATX-II ATX-III ATX-V	4000 100 18,000 19.0	4.4 3.7 6.7 10.4	7.0 0.15 10.0 0.05	[21,23,26,97,153]
*A. xanthogrammica*	ApA ApB	66.0174 8.0	22.0 78.0	0.12 0.035	[30] [25,26]
*S. gigantea*	Gigantoxin II	2000	14.0	10.0	[26,103]
*R. paumotensis* (*=H. magnífica*)	RpI RpII RpIII RpIV	145 4200 53.0 40.0	36.0 15.0 10.0 90.0	0.9 10.0 0.3 10.0	[27] [27] [91] [27]
*S. helianthus*	ShI	20,000	0.6		[31,118]
*P. actinostoloides*	PaTX	20,000	10.0		[126,130,134]
*H. crispa*	RTX-I RTX-II RTX-III RTX-IV RTX-V δ-SHTX-Hcr1f	3000 1650 25.0 40.0 350 4200	3.5 4.0 82.0 4.4 12.0 15.0		[28] [115] [29] [116] [116] [168]

^a^ Lethal doses for mice when injected intraperitoneally. ^b^ The equilibrium dissociation constant of the resulting complex of neurotoxins with rat brain synaptosomal Navs.

**Figure 8 toxins-15-00008-f008:**
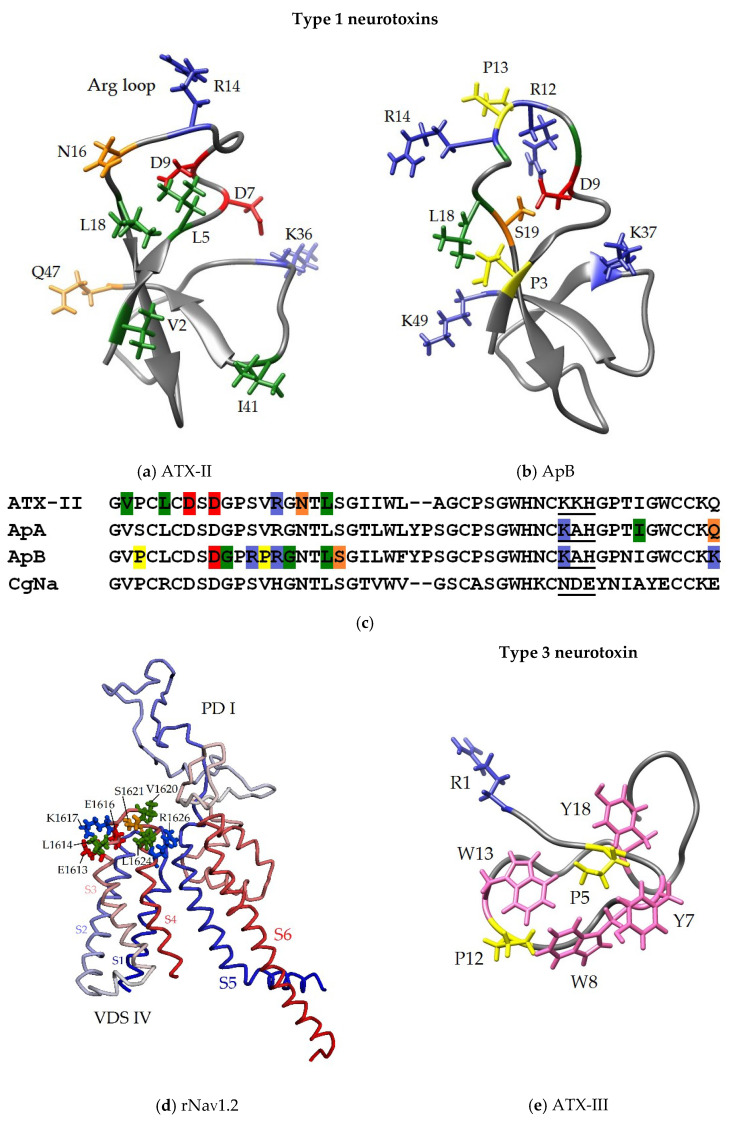
Functionally significant amino acid residues of NaTxs (**a**–**c**,**e**) and ATX-II putative binding site (**d**). The homology models of ATX-II (based on ApA, PDB ID: 1Ahl) (**a**), and PD-I with VSD-IV of rNa_V_1.2 channel (based on hNa_V_1.2, PDB ID: 6J8E) (**d**), and 3D structure of ApB (1Apf) (**b**), and ATX-III (1ANS) (**e**). The ribbon diagram of the neurotoxins is shown, key residues presented as sticks and colored blue (basic), red (acidic), green (aliphatic), orange (polar), yellow (proline) and pink (aromatic). The ribbon diagram of rNa_V_1.2 fragment is colored from N- (blue) to C-termini (red), key ATX-II interacting residues are colored as described above. (**c**) Multiple alignment of the amino acid sequences of the type 1 NaTxs; key residues are colored as described above, and positively (ATX-III, ApA, ApB) or negatively (CgNa) charged motif is underlined.

In addition, it was found that positively charged residues of ApA and ApB caused a difference in the binding affinity of those neurotoxins to Na_V_s, and Arg12, Leu18, and Lys49 [157,158,165] (Figure 8c) were the most significant for binding ApB with Na_V_s and for cardiospecificity. A structure-functional analysis of the ApB residues showed that besides one of the two positively charged C-terminal residues, Lys49, such residues as Asp9 and Lys37 (Figure 8b,c) were extremely important for the formation of this neurotoxin complex with Na_V_ [147]; they provided electrostatic interaction of the neurotoxin charged residues with oppositely charged residues of channel [62,167]. Thus, it was found that the binding of ATX-II to the rat Na_V_1.2 channel involved its charged residues, Glu1613, Glu1616, and Lys1617, as well as the hydrophobic residues, Leu1614, Val1620, Ser1621, Leu1624, and Arg1626, located on the domain DIV linker S3–S4 [121] (Figure 8d), while Lys37 of ApB interacted with Asp1612 of the rNa_V_1.5 channel [62]. It is believed that the Na_V_ residues that form binding site for NaTxs of 1, 2, and 3 types may also differ [22].

It has been shown that most NaTxs types 1 and 2 have positively charged residues within the neurotoxin ‘Arg14 loop’, which is of great importance for the formation of a complex with the Na_V_ channel [147]. This loop is assumed to be the one that provides binding affinity of NaTxs with Na_V_s site 3 and their selectivity in relation to different channel subtypes [123]. The presence of at least one positively charged residue in this loop (mainly Arg14) is necessary not only for the binding of a neurotoxin to a sodium channel, but also for its subsequent stabilization in conducting state [164,165]. However, the study of the functional role of a very conservative Arg14 residue detected ambiguous results of its chemical modification [147]. Therefore, the ApB mutagenesis showed less significance of Arg in the 14 position, compared to that of Arg in the 12 position. NaTxs, such as PCR2-1, PCR2-5, PCR3-6, Am-III, CgNa, CGTX-III, Rc-1, Cp1, Cp2, Hk7a, and AdE-1, were characterized by the absence of positively charged residues in ‘Arg loop’. At the same time, these neurotoxins had one Arg residue per loop which occupied only position 5 (Figure 2). It is quite obvious that a role of Arg14 in the type 2 neurotoxin molecules lacking Arg12 may be more significant, since they contain only one positively charged residue in this loop, Arg13 [161] (with the exception of Arg13 in RTX-VI [168]). It was also found that the toxicity to mice of the type 2 neurotoxin ShI [31], despite the presence of Arg4 and Arg13, was reduced significantly due to the increase of a negative charge at N-terminus (a cluster with Asp6, Asp7, Glu8, and Asp11, which is not typical for the activity of the type 1 neurotoxins, ApB and ATX-II) [37,163]. At the same time, the functionally crucial residues of type 1 neurotoxins were not conservative for type 2 [37]. The authors suggested that further study and analysis of the structure-functional relationships between NaTxs and Na_V_s would make it possible to establish the selectivity of their modulating action on all subtypes of these channels [37].

The comparative structure-functional analysis of ApB, ATX-II, and CgNa suggested that the cluster of negatively charged residues, Asn35-Asp36-Glu37, in the CgNa molecule (Figure 8c) reduced the binding affinity of the neurotoxins to Na_V_ channels [81,83,147,163,169]. It should be emphasized that the residues in these positions of a number of other NaTxs types 1 and 2 were mostly positively charged or neutral.

It has been supposed that NaTxs may bind to their target channels not only through charged residues, but via the hydrophobic interaction of aliphatic and aromatic amino acid residues localized on the molecular surface [37]. Thus, mutagenesis of the ATX-II residues has demonstrated that identical hydrophobic amino acids (Val2, Leu5, Asn16, Leu18, and Ile41 (Figure 8a,c), are involved in an interaction with mammals and insects Na_V_s [163]. Moreover, according to the results of scanning mutagenesis, hydrophobic residues Pro3, Pro13, highly conserved Gly10, Gly15 and Leu18-Ser19 (Figure 8b,c) belonging to the ApB ‘Arg loop’, provide conformational flexibility of the molecule, while differences in the value of modulating activity and cardioselectivity of ApB [147,165,170] are associated with the substitutions Ser12Arg, Gln49Lys, as well as Ser3Pro and Val13Pro, respectively, which makes the structure of the ‘Arg loop’ more rigid [37].

From the point of view of the specific NaTxs residues, functional importance for Na_V_ modulating activity and toxicity, ApB [144,147,164,165,171] and ATX-II [163] are the most studied. Scanning mutagenesis revealed differences between their molecular surfaces, which, in turn, were due to different surface localization of functionally significant residues that determined the direction, magnitude of the dipole moment (Figure 7) and the orientation of neurotoxin molecules in complex with the channel [81]. Thereby, the functionally significant ATX-II residues somewhat differed from those of ApB (Figure 2 and Figure 8).

The use of ApB, along with other venomous toxins, in particular, tarantula huwentoxin-IV (HWTX-IV), has shown that the binding of a neurotoxin to receptor site 3 stabilizes voltage-sensitive domain VSD-IV in an inactivated state, while to open Na_V_ it is enough to activate domains VSD-I–VSD-III [172]. Despite the lack of homology, NaTxs were observed to compete with scorpion α-toxins for binding to Na_V_ receptor site 3, and it was found that they could also act synergistically with toxins binding to site 4 [37]. In addition, it was shown that α-scorpion toxin derivatives were covalently attached to amino acid residues located in a DI extracellular loop between the transmembrane helices S5 and S6 [60].

Although binding site 3 is common for scorpion α-toxins and spider δ-toxins, which have a similar effect on channel currents, the results of the mechanical and electrophysiological study indicate that the binding sites of various neurotoxins with Na_V_s rather overlap than are identical [121]. The authors believe that the Na_V_ residues that form binding sites for NaTxs types 1, 2, and 3 can also differ [110,122].

The main functionally significant NaTx residues of types 1 and 2, which determine the affinity and specificity of neurotoxins for certain Na_V_ types of arthropods and/or mammals, differ significantly from those of type 3 neurotoxins, ATX-III (=Av3) [122] and PaTX [134]. This is stipulated by differences in the amino acid sequence and the spatial structure of three NaTx types (Figure 2, Figure 3 and Figure 4, Figure 7 and Figure 8) [129]. It has been shown that N-terminal positively charged Arg1 as well as several hydrophobic residues, Pro5, Pro12, Tyr7, Tyr18, Trp8, and Trp13 (Figure 8e), are functionally significant residues for ATX-III toxicity (Table 1) [122]. At the same time, the toxicity of other type 3 neurotoxins structurally different from ATX-III, in particular PaTX [173], is stipulated by the presence of two charged residues, Lys4 and His27, as well as by three hydrophobic ones, Tyr15, Pro20, and Trp21.

Unlike NaTx (types 1 and 2) and scorpion α-toxin, which interact with external S1–S2 linkers and S3–S4 linkers on the DIV voltage sensor and with neighboring external linkers on the DI pore module, neurotoxin Av3 (type 3) interacts with the DI linker/SS2–S6 adjacent to the channel pore. According to the data of point mutagenesis and structure-functional analysis, Av3 hydrophobic residues localized on the molecular surface are involved in the neurotoxin-channel interaction. They can bind to the Trp404 and to His405 residues of the insect Nav at a cleft near the S6 segment of the DI domain [60]. It has been shown that substitutions of positively charged residues (R4E and R5E) in the S4 segment of DIII, as well as negatively charged residues (D1K and E2K) in the S2 segment of the same domain, lead to an increase in the activity of the site-3 and site-4 toxins, LqhαIT from *Leiurus quinquestriatus hebraeus* and Av3 from *A. viridis*, selective to insect sodium channels [169].

The toxicity of the most studied neurotoxins of types 1, 2, and 3 are in a wide range of values, as well as of species and tissue specificity (Table 1), which may be due to differences in the surface properties of their molecules. Generally, different venomous toxins modulating Na_V_ channels possess more than one mechanism of action, and they can be seen as gating modifiers [148]. However, currently available experimental data cover only a limited number of sea anemone neurotoxins. To establish their critical amino acid residues involved in the interaction with various types of Na_V_ channels, it is necessary to expand a NaTxs arsenal significantly [169].

Finally, until now, there has been no clear correlation between the neurotoxin fold, the scaffold of disulfide bridges, and Na_V_ modulating activity. The folding and organization of disulfide bridges play a crucial role in the spatial distribution of residues, which are key to the pharmacology of the toxin. Sometimes, the recognition of ion channel subtypes (selectivity, affinity, activity) by neurotoxins can be achieved through minor or more marked changes in the structures of NaTxs (for example, in solvent-exposed residues as well as in the direction of the dipole moment) [81,168].

### 3.4. Toxicity of NaTxs

One of the most interesting features of types 1 and 2 NaTxs is the variability of their toxicity to crustaceans and mammals. In vivo NaTxs testing in mammals (mice) and arthropods (crabs) indicates that their very high toxicity for crabs varies much less (20 times) than that for mice (150 times) (Table 1). At the same time, there is a correlation between the toxicity of neurotoxins on these two organisms. NaTxs more toxic to mammals are less toxic to arthropods, and vice versa.

It is shown that the substitution of Ser12Arg and C-terminal Gln49Lis in an ApB sequence leads to a 4-fold increase in the binding constant of a neurotoxin to Nav synaptosomes in a rat brain, compared to ApA [30]. The fact of great interest is that two type 2 toxins, RTX-III and ShI, differ significantly in toxicity to mammals, although the degree of their homology is very high (88%). Comparing sequences, substitutions at the positions Asp11, Thr20, and Ser25 in ShI with the Tyr residue in RTX-III (Figure 3) (with the exception of the conservative substitutions Ile12Val, Thr38Ser, Asp42Glu) are of huge interest. In this case, the Tyr residue appears to be at position 25 in ApA and ApB (Figure 2), the most toxic for mice. It is obvious that the substitution of Ser residue at these positions with Tyr (RTX-I, RTX-III–RTX-V) (Figure 3) leads to an increase in the toxicity of neurotoxins for mammals (Table 1). Thus, the values of the lethal doses of *Heteractis* neurotoxins for mice varied from 3000 μg/kg for RTX-I, 1650 for RTX-II, 350 for RTX-V, to 25 and 40 μg/kg for RTX-III and RTX-IV, respectively, while for crabs the opposite picture was observed.

The most toxic peptides for mammals are ApB, ATX-V, and RTX-III (Table 1). It has been shown that the RTX-III action on the neuromuscular transmission is similar to the effect of ATX-II and ApB [65,93,174]. Nevertheless, fulfilling their ecological function in biocenosis, almost all representatives of the studied types are the most toxic for arthropods; their lethal doses for crabs being in the range from 0.6 (ShI) to 90.0 (RpIV) of values.

RTX-III and RTX-IV differing only in two substitutions, Tyr11Asn and Glu31Asp, have practically the same LD_50_ values, and a decrease in the amount of Tyr residues to one in RTX-I leads to a 150-fold decrease in its activity (Figure 3, Table 1). The analysis of residues in the sequences and the toxicity of type 2 representatives shows that RTX-III, the most toxic for mice, contains the largest amount of Tyr residues (5), and does not contain Phe residues. At the same time, RTX-I and RpII, the least toxic to mammals, have two Phe residues (Figure 3, Table 1).

There are also some differences in the amount of residues Glu, Gln, Ala, and Thr in neurotoxin sequences. It should be noted that RTX-III and RTX-IV, comparable in activity, differ in the substitution of only one Tyr/Glu or Tyr/Gln (Figure 3). Probably, the high toxicity of RTX-III in mammals is also associated, in addition to a high positive charge at C-terminus and negative charge at N-terminus, with the substitution of negative Asp11 with an aromatic Tyr residue, which can increase the affinity with a target due to π-π interactions, which, according to [81], arise between their aromatic residues [168]. Toxicity of δ-SHTX-Hcr1f in mice [168] was observed to be consistent with data having previously been reported for RpII from *H. magnifica* (LD_50_ 4200 μg/kg) [27,175].

According to the results of binding experiments with some NaTxs and rat brain synaptosomes (Table 1), the equilibrium dissociation constant of the complex (K_D_) differ by three orders of magnitude in the range from 0.035 (ApB) to 10.0 μM (ATX-III, Gigantoxin II, RpII, and RpIV).

### 3.5. Evolution of Sea Anemone Neurotoxins

In recent years the intensive genomic, transcriptomic, and proteomic investigations of structure-function, phylogenetic, molecular-evolutionary relationships of sea anemone neurotoxins, and their encoding genes, have been carried out [14,17,18,19,37,95,136,152,176]. Some authors found that, despite a long evolution over 600 million years, most of the characterized neurotoxins belonging to different structural types had strongly been influenced by a negative (purifying) selection reducing their genetic diversity [18,37] and ensuring high conservation of neurotoxin structures (with the preservation of functionally significant Arg14 [14]) and functions necessary for their specificity to different Na_V_ subtypes [152].

These sea anemone evolutionary processes are in a sharp contrast with those of younger lineages, such as cones, snakes, and scorpions, whose genes are much more receptive to a diversifying (positive Darwinian) selection [37,177,178,179,180,181] leading to the genetic diversity of encoded toxins as emphasized by the authors [95]. In addition, it was found that NaTx residues located on the surface of molecules could be influenced by a positive selection. (Thus, types 1 and 2 neurotoxins modulating Na_V_ as well as type 3 K_V_ toxins have a common evolutionary origin, and their different pharmacological properties are the result of a rapid adaptive evolution [14]).

Moran et al. have found that type 1 NaTxs produced by *N. vectensis*, as well as some other species, are encoded by gene families with a very similar genomic organization [37,95]. The analysis of the nucleotide sequences of genes encoding *N. vectensis*, *A. viridis*, and *A. equina* neurotoxins has shown their surprisingly high conservation within each of the species which is the result of a concerted evolution aimed at increasing the level of a gene expression.

Using comparative genomics and transcriptomics to study the inter- and intraspecific variability of venom toxins under strong selective pressure and its influence on expression, allowed deep analysis of a number of regularities in the expression of many toxic proteins [182]. The authors found that the venom phenotype of sea anemones of each species may be determined by the rapid and dynamic expression of the gene family of the dominant toxin through duplication. It has been established that changing environmental conditions affect micro- and macroevolutionary models of changes in the phenotype of sea anemone venom within and between species. At the same time, it was found that the number of copies of the dominant toxin (in particular, the model neurotoxin Nv1 from *N. vectensis*) can vary significantly in different populations. This is consistent with different Nv1 expression (at the transcript and protein levels). According to the results of phylogenomic analysis, gene duplication is the main mechanism that explains adaptive shifts in gene expression.

Despite the high amino acid sequences conservation of NaTxs, some species of sea anemones produce from 2–3 to more highly homologous representatives [36]. It should be noted that the currently insufficient arsenal of neurotoxins with the fully established nucleotide sequences of coding genes slows down the progress in the study of genes families and their evolutionary relationships [95].

## 4. Neurotoxins from Sea Anemone *H. crispa* (=*R. macrodactylus*)

### 4.1. Isolation, Modifications, Amino Asid Sequences 

One of the first sea anemone species, the study of which was started in the late 1970s, is the tropical sea anemone *Radianthus macrodactylus* (=*Heteractis crispa*). From its water-ethanol extracts by hydrophobic chromatography on Polychrome-1, ion-exchange chromatography on Bio-Rex 70, SP-Sephadex C-25, and reversed-phase HPLC on Silasorb C18, five individual neurotoxic peptides (RTX-I–RTX-V) were isolated [28,29,115,116]. Full amino acid sequences determined using direct peptide sequencing by the Edman degradation method allowed the authors to classify the peptides as type 2 neurotoxins (Figure 3). The spatial ^1^H NMR structure of RTX-III, determined by Hinds and Norton [183], was similar to the another type 2 neurotoxin, RpII [27].

The first systematic analysis of the structure-functional relationships of type 2 neurotoxins, as well as an attempt to elucidate the mechanism of their interaction with targets, were undertaken with RTX-III [28,29,84,115,116,160,161,184,185,186,187,188,189,190,191]. A series of experimental works was carried out on the chemical modifications of some RTX-III amino acid residues (Table 2) [188,189,190,191]. A significant decrease in activity (12 times lower) was observed as a result of the modification of the N-terminal ε-amino group of Gly1, while the modification of other residues affected the activity of RTX-III to a lesser extent and, according to the data of CD spectroscopy, was not accompanied by a change in the conformation of a molecule [161]. It was found that the protonated carboxyl and amino groups of some residues were important for the manifestation of the functional activity of RTX-III.

The obtained data shown in Table 2 suggested the absence of the critical residues in the RTX-III sequence, which were absolutely necessary for neurotoxins functional activity [188,189,190,191]. These data of Mahnir et al. contradicted the classical concept of an ‘active site’ that existed in the early 1970s, as well as the previously proposed model of organizing a reactive site with an Arg13 residue as a receptor site for neurotoxin binding to the Na_V_ channel [162]. The results obtained showed that the responsibility for the toxic properties of neurotoxins was distributed between many functional groups of amino acid residues, among which the amino groups and the guanidine group of Arg residues capable of carrying a positive charge in the protonated state were distinguished. Numerous next studies showed that an ‘Arg loop’ is crucial for neurotoxins affinity and selectivity toward Na_V_ channels [110,162]. The change in neurotoxin activity upon the modification of charged residues indicated that the RTX-III binding to the NaVs was largely determined by electrostatic interaction and, most likely, had a multipoint character [84,187].

Thus, the modification of only one or two functionally important charged residues (in particular Arg, Lys) only slightly changes the functional activity of residues remaining free (Table 1) [188,189,190,191]. At the same time, according to the CD spectroscopy data, which showed an insignificant change in the spatial structure of RTX-III upon the destruction of disulfide bridges by 2-mercaptoethanol, there was an almost complete drop in the activity of the neurotoxin (100 times) [184]. This indicated that disulfide bonds were important for maintaining the active conformation of the RTX-III molecule.

### 4.2. New Double-Stranded Type 2 Heteractis Neurotoxin

Recently, two NaTxs have been isolated and identified from *H. crispa* by chromatographic methods and the combination of sequencing by automated Edman degradation, a proteomic analysis, and tandem mass spectrometry [168]. One of peptides, namely δ-SHT-Hcr1f, according to its molecular mass (5287 Da), amino acid sequence (Figure 3), and toxicity (Table 2), has been identical to the RpII neurotoxin, which was first isolated from the sea anemone *R. paumotensis* and previously described by Schweitz et al. [27]. The second peptide, RTX-VI (5240 Da), was identified as the new representative of *Heteractis* neurotoxins. The determination of its amino acid sequence resulted in the unusual structure of a double-stranded analog of RTX-III (5378 Da) [29] lacking the Arg13 residue (Figure 9).

Two peptide chains of the RTX-VI molecule (12 and 35 aa) are linked, according to RTX-III sequence, by two disulfide bonds, C3-C43 and C5-C33. Thus, the third disulfide bridge binds the residues Cys26 and Cys44 localized in chain 2 (Figure 9). Such a sequence unique for sea anemones peptides was first discovered by the authors among the representatives of NaTx type 2 and was confirmed by the methods of CD spectroscopy and molecular modeling [168].

A comparative analysis of RTX-III and RTX-VI 3D models of spatial structures with various alternative connections of Cys residues in the disulfide bond (obtained by homologous modeling with the ShI as prototype model) showed that only one of several RTX-VI theoretical models satisfied the β-defensin-like fold, which is characteristic of NaTx types 1 and 2 with disulfide bridges pattern C1-C5, C2-C4, and C3-C6 [168] (Figure 9). The authors suggested the existence of a post-translational modification of the RTX-III molecule, which could lead to its cleavage and removal of Arg13, resulting in the emergence of a double-stranded RTX-VI molecule [168]. A similar phenomenon of polypeptide chains cleavage and formation of double-stranded molecules was previously found for *Cupiennius salei* conus toxins during their post-translational modification [192]. While RTX-III exhibits very high toxicity (LD_50_ 25 µg/kg, Table 1), there is no toxicity in the case of RTX-VI (LD_50_ > 10 mg/kg) [168].

## 5. Interaction of NaTxs with NaV Subtypes Tested Electrophysiologically and by Molecular Docking

Among type 1 NaTxs subjected to electrophysiological studies, the most known toxins are ApA and ApB (*A. xanthogrammica*), ATX-I–ATX-III, ATX-V (*A. sulcata*), AFT-II (*A. fuscoviridis*), BcIII (*B. caissarum*), CgNa (*C. gigantea*), BgII and BgIII (*B. granulifera*), CGTX, CGTX-II, and CGTX-III (*B. cangicum*) [2,35,80,90,110,166,167]. The neurotoxins ATX-II, ATF-II, CgNa, BcIII, and CGTX-II have been characterized most completely in terms of selectivity towards various isoforms of Na_V_ in mammals (Table 3) [193,194].

The electrophysiological testing of *Heteractis* δ-SHTX-Hcr1f, RTX-III, and RTX-VI neurotoxins on nine cloned voltage-gated sodium channels, has shown that they modulate the BgNa_V_1 subtype of insects, the VdNa_V_1 of arachnids, selectively affect mammalian CNS Na_V_1.1–Na_V_1.3, and Na_V_1.6 subtypes, but are not active towards skeletal muscle Na_V_1.4, cardiac Na_V_1.5, and PNS Na_V_1.8 subtypes (Table 3) [168]. None of the *H. crispa* neurotoxins, unlike the type 1 ones, affect the Na_V_1.5 currents of cardiomyocytes. According to sodium channels localization in an organism, most diseases associated with Na_V_ channelopathies affect nervous, cardiovascular, or musculoskeletal systems [2,40,63,71,72,73,74,92]. Previously, Schweitz and co-authors found that RpII isolated from *H. magnifica* (=*H. paumotensis*) slowed down the inactivation of tetrodoxin-sensitive Na_V_ currents expressed by neuroblastoma cells (NIE-115) and tetrodotoxin-resistant Na_V_ currents in rat skeletal muscle tissue myoblasts [27] The results of Kalina et al. [168] confirmed that δ-SHTX-Hcr1f (=RpII) activated Na_V_1.2, the major subtype of channels expressed by neuroblastoma cells. However, δ-SHTX-Hcr1f did not have a modulating effect on the currents of the tetrodoxin-resistant channels Na_V_1.5 and Na_V_1.8 (Table 3).

In general, δ-SHTX-Hcr1f, RTX-III, and RTX-VI are significantly more selective for mammalian channels Na_V_1.1–Na_V_1.3, than type 1 neurotoxins BcIII, ATX-II, and ATF-II. The pharmacological profiles and observed effect of *Heteractis* neurotoxins on the subtypes Na_V_1.1–Na_V_1.3, Na_V_1.6, BgNa_V_1 and VdNa_V_1 correlate with those of NaTxs types 1 and 2 from some sea anemone species (Table 3) [2,27,59,63,80,168].

According to data of Kalina et al. [168], and data presented by Olivera et al. [59], the effect of these neurotoxins on different Na_V_s subtypes is manifested at significantly varied values of the half maximal effective concentration (EC_50_) (Table 4).

Thus, *Heteractis* neurotoxins δ-SHTX-Hcr1f, RTX-III, and RTX-VI, effectively potentiate insect and arachnid channels (with high EC_50_ values) and less effectively potentiate mammalian channels (with low EC_50_ values). This is not surprising, as many studies have established the varied bioactivity of different functionally significant amino acid residues, which is stipulated by their certain localization on the surface of a molecule and its physicochemical characteristics.

The absence of the Arg residue at position 13 in RTX-VI (equal to Arg14 in most NaTxs and highly conserved for all, according to [2,163]) and the cleavage of its polypeptide chain, change the activity profile of this neurotoxin in relation to different Na_V_ subtypes (Table 3). A double-stranded peptide becomes unable to modulate Na_V_1.3, however, despite reduced efficiency and unlike RTX-III, the neurotoxin RTX-VI activates Na_V_1.2. At the same time, the activity of RTX-III and RTX-VI against seven other Na_V_ subtypes remains almost identical (Table 3). The authors believe that the absence of Arg13 (functionally important for the affinity and the selectivity of all NaTxs) can be partially compensated by positively charged residues at RTX-VI C-terminus important for the interaction with Na_V_1.2, Na_V_1.6, BgNa_V_1 and VdNa_V_1 subtypes. Most likely, Arg13 is responsible for the molecular recognition and the selectivity of the interaction with Nav isoforms and is not functionally significant for channel binding. This is in a good agreement with the results of the chemical modification of Arg13 in RTX-III (Table 2), leading to a drop in its toxicity, and with the hypothesis of the ‘multipoint’ interaction of NaTxs with Na_V_ [84,161,185,186,187] and consistent with Khera and Blumenthal data [164,165], which show that Arg14 in ApB (equal to Arg13 in RTX-III) is less important for the conservation of the neurotoxin activity than Arg12.

An in silico study of the δ-SHTX-Hcr1f interaction with Na_V_1.2 has shown that the channel binding site with this neurotoxin is located not only in the extracellular region of the VSD-IV domain whose Glu1616, Lys1617, and Val1620 residues interact with charged neurotoxin residues Asp6, Arg13, and Lys32 (while positive residue Lys37 of ApB interacts directly with negative charged Asp1612 of rat Na_V_1.5 [62]). This Na_V_1.2 binding site with δ-SHTX-Hcr1f also belongs to the extracellular region of the pore-forming PDI domain, whose Asp317, Gly318, Lys355, Asn285, and Asn340 residues interact with Asp7, Glu31, Arg45, Lys46, and Lys48 of the neurotoxin by means of the network of electrostatic interactions and hydrogen bonds [168]. Recently, similar binding of sea anemone neurotoxins Av1, Av2, Av3, and CgNa to a number of the PDI domain residues of the cockroach Na_V_PaS channel through the formation of hydrogen bonds and salt bridges has been found [87].

Thus, molecular dynamics simulations data of the δ-SHTX-Hcr1f–rNa_V_1.2 complex demonstrated that, by binding to the channel, the neurotoxin caused significant conformational changes in the extracellular region of the pore-forming PDI domain. At the same time, the observed weaker interaction with the S3–S4 loop hindered the conformational movement of the S4 helix relatively to the other helices of the VSD-IV domain which prevented the channel transition from an activated to inactivated state. In addition, the MD simulation of the δ-SHTX-Hcr1f–rNa_V_1.2 complex also revealed the set of hydrophobic and electrostatic contacts with polar head groups of cell membrane phospholipids adjacent to the VSD-IV domain [168]. These in silico data are confirmed by electrophysiological tests.

According to the authors’ data, RTX-VI, RTX-III, and δ-SHTX-Hcr1f, are the less potent modulators of insect Na_V_ channels, comparing to the mammalian channels. All three *Heteractis* peptides, in most cases, significantly increase the value of membrane potential required for the development of both activation and inactivation of the channel inhibiting both processes (Figure 10). They are able to shift the inactivation curves in the direction of positive values of the membrane potential, i.e., make the inactivation of the channel incomplete and less sensitive to potential changes, which is typical for the previously described sea anemone toxins and scorpion α-toxins. Like most sea anemone neurotoxins, δ-SHTX-Hcr1f, RTX-III, and RTX-VI, being Na_V_ activators, amplify both the peak and tail components of currents and slow down the kinetics of Na_V_s inactivation [168].

It has been established that neurotoxins belonging to structural type 2 do not have pronounced species specificity, acting in nanomolar concentrations on the currents of voltage-gaited Na_V_ channels in mammals and arthropods; but they interact with several CNS-specific mammalian Na_V_ subtypes and stabilize the channels in an open state, inhibiting their kinetic inactivation. 

Further studies of the molecular mechanisms of NaTxs interactions with Na_V_s will certainly correct and deepen the current understanding of intermolecular contacts in the formed three-molecular complexes of neurotoxins with targeted channels and membrane phospholipids.

## 6. Conclusions

The data presented in this review have been obtained over the past four decades as a result of structural, chemical, biochemical, electrophysiological, molecular biological, and in silico studies. We have described some data on the classification, the amino acid sequences, the spatial structures of known sea anemone neurotoxins belonging to structural types 1–4, as well as data on several new neurotoxins not related to given structural types. We have also summarized data on six representatives of type 2 NaTxs produced by the tropical sea anemone *H. crispa*, including a double-stranded peptide RTX-VI first discovered in sea anemones. This peptide is the analog of the most toxic *Heteractis* neurotoxin, RTX-III, but it does not have Arg13 residue, which results in the loss of its toxicity. The results of electrophysiological testing of δ-SHTX-Hcr1f, RTX-III, and RTX-VI, have shown the presence of the affinity binding with different types of sodium channels. This review has presented the calculated data first obtained by molecular modeling for the prediction of protein–protein and protein–lipid binding interactions of the *Heteractis* neurotoxin δ-SHTX-Hcr1f with Na_V_1.2, and with phospholipids surrounding the channel.

The sea anemone neurotoxins have huge potential as valuable molecular biochemical tools for studying the structural organization and the functioning mechanisms of various Na_V_ subtypes, as well as being promising pharmacological agents encouraging a large-scale search for new representatives, using the traditional methods of proteomic, gene cloning, and high-throughput transcriptomic sequencing, with the aim of further design of low-toxic or non-toxic pharmacological peptides on their basis.

## Figures and Tables

**Figure 1 toxins-15-00008-f001:**
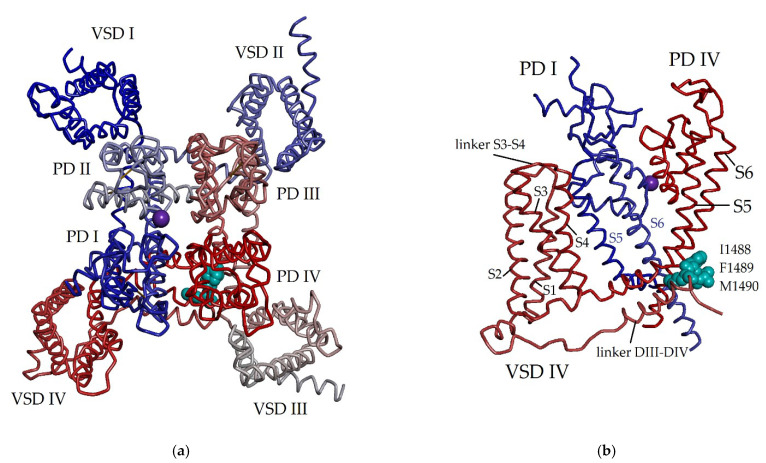
The ribbon diagram of hNa_V_1.2 channel (PDB ID: 6J8E) (**a**) and receptor site 3 with the DIII−DIV linker, responsible for fast inactivation (**b**). The channel is colored from N- (blue) to C-termini (red); the inactivation particle—IFM-motif, and sodium ion are presented as cyan and violet CPK, respectively.

**Figure 6 toxins-15-00008-f006:**
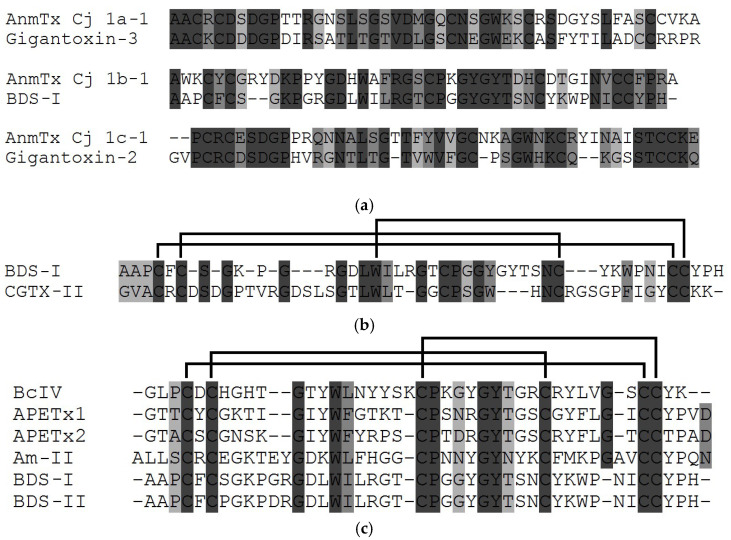
Amino acid sequences of: (**a**) AnmTX Cj 1a-1, AnmTX Cj 1c-1, AnmTX Cj 1b-1, gigantoxin-3, gigantoxin-2 from *S. gigantea* [103,139], and BDS-I (UniProt ID: P11494) from *A. sulcata* [140]; (**b**) BDS-I and CGTX-II from *B. cangicum* [106]; and (**c**) structurally new class of so-called BcIV- and APETx-like peptides [107]: BcIV (UniProt ID: P84919) from *B. caissarum* [141], APETx1 (P61541) [41] and APETx2 (P61542) [42] from *A. elegantissima*; Am-II (P69930) from *A. maculata* [100]; BDS-I and BDS-II (P59084) [142] from *A. sulcata*. Hydroxy-Pro at position 24 of Am-II is denoted by ‘O’. The sequence similarity is shown as a dark (high) and light (low) gray background, the multiple sequence alignment was performed using the Vector NTI software.

**Figure 9 toxins-15-00008-f009:**
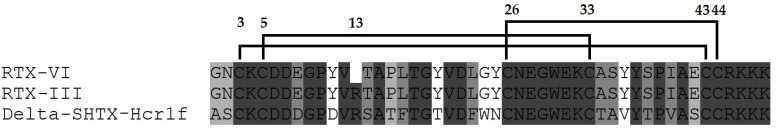
Amino acid sequences of the *Heteractis* neurotoxins: RTX-VI, RTX-III, and δ-SHTX-Hcr1f (=RpII). Amino acid sequence of RTX-VI includes two peptides (residues 1–12 and 14–48), both peptides are connected by two disulfide bonds, C1-C5, C2-C4. The disulfide bonds are shown above the alignment. Identical residues are shown in grey.

**Figure 10 toxins-15-00008-f010:**
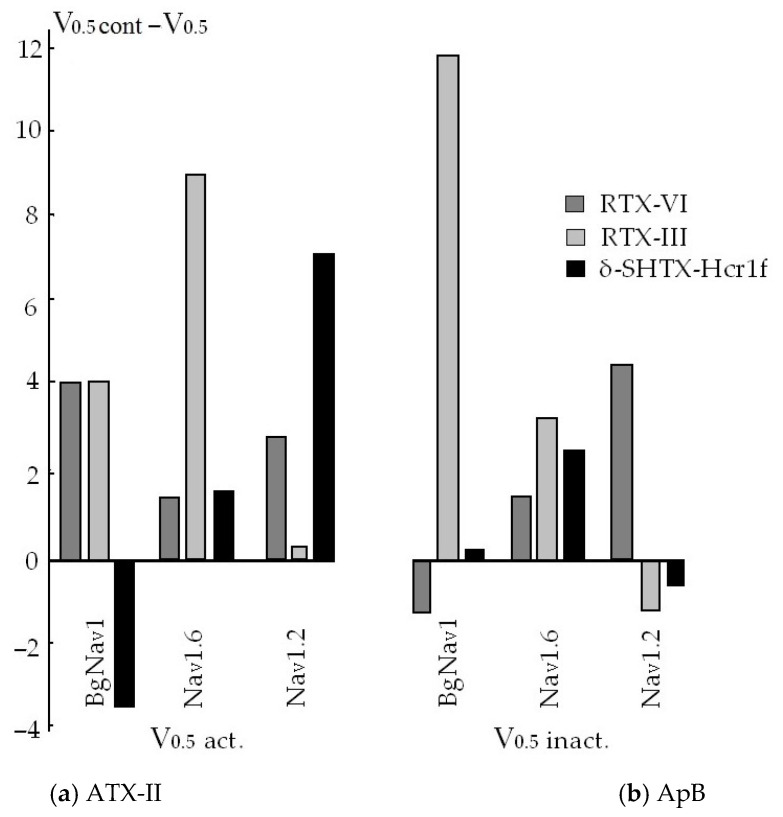
A change in the membrane potential value, at which the activation/inactivation of the channel reaches 50% (V_0.5 act._ and V_0.5 inact._) with the presence of 100 nM of *Heteractis* neurotoxins: (**a**) the inhibition of activation; (**b**) the inhibition of inactivation.

**Table 2 toxins-15-00008-t002:** Effect of chemical modification on RTX-III toxicity for mice [160,161,188,189,190,191].

Modified Residue(s)	Reagent	LD_50_ Modif./LD_50_ Nativ (Mice)
Gly1 (α-NH2) Lys4 Lys (C-terminal)	[^3^H]-acetic anhydride	12 2 2
Arg13	malonic aldehyde phenylglyoxal 1,2-cyclohexanedione	4–5
COOH	[^3^H]-glycine methyl ether/EDC [^14^C]-trimethyloxonium tetrafluoroborate	1–2
Trp30	2-hydroxy-5-nitrobenzyl bromide	1
S-S bonds	2-mercaptoethanol/iodoacetamide	>100

**Table 3 toxins-15-00008-t003:** Effects of neurotoxins BcIII, ATF-II, ATX-II [59], CGTX-II, δ-AITX-Bcg1a, δ-AITX-Bcg1b [155], CgNa [156], Nv1, Nv5, Nv6 [17], BgII, BgIII [81,105], RTX-III [29,168], δ-SHTX-Hcr1f, and RTX-VI [168] on the Na^+^ currents of mammals Na_V_1.1–Na_V_1.8, insect BgNa_V_1, and arachnid VdNa_V_1 channels expressed in *Xenopus laevis*.

NaTx	Voltage-Gated Sodium Channel Subtype									
	Na_V_1.1	Na_V_1.2	Na_V_1.3	Na_V_1.4	Na_V_1.5	Na_V_1.6	Na_V_1.7	Na_V_1.8	BgNa_V_1	VdNa_V_1
δ-SHTX Hcr1f	+	+	-	-	-	+	n.d.	-	+	+
RTX-III	-	-	+	-	-	+	n.d.	-	+	+
RTX-VI	-	+	-	-	-	+	n.d.	-	+	+
BcIII	+	+	+	+	+	+	n.d.	n.d.	n.d.	n.d.
AFT-II	+	+	+	+	+	+	n.d.	n.d.	n.d.	n.d.
ATX-II	+	+	+	+	+	+	n.d.	n.d.	n.d.	n.d.
CGTX-II	-	-	-	-	+	+	-	n.d.	n.d.	n.d.
δ-AITX-Bcg1a	+	+	+	+	+	+	-	n.d.	n.d.	n.d.
δ-AITX-Bcg1b	-	-	-	-	-	-	-	n.d.	n.d.	n.d.
BgII	n.d.	+	n.d.	+	+	n.d.	n.d.	-	n.d.	n.d.
BgIII	n.d.	+	n.d.	+	+	n.d.	n.d.	-	n.d.	n.d.
CgNa	-	-	+	-	+	+	-	n.d.	n.d.	n.d.
Nv1	-	-	-	-	-	-	n.d.	-	+	n.d.
Nv4/Nv5	-	-	-	-	-	-	n.d.	-	-	n.d.

n.d.—The activity of neurotoxins in relation to Na_V_ channel subtypes was not determined.

**Table 4 toxins-15-00008-t004:** The half-maximal effective concentration (EC_50_ nM) of *Heteractis* neurotoxins, δ-SHTX-Hcr1f, RTX-III, RTX-VI [168], and ones of type 1, ATX-II, AFT-II, BcIII, BgII, BgIII δ-AITX-Bcg1a, CGTX-II [17,59,105,155,156], under the action on sodium channel subtypes, Na_V_1.1–Na_V_1.7 and BgNa_V_1.

Toxin	EC_50_ nM							
	Na_V_1.1	Na_V_1.2	Na_V_1.3	Na_V_1.4	Na_V_1.5	Na_V_1.6	Na_V_1.7	BgNa_V_1
SHTX-Hcr1f		79.5				183.5		226.1
RTX-III		–				381.8		978.1
RTX-VI		120.9				282.3		760.5
ATX-II	6.01	7.88	759.22	109.49	49.05	~180	1800	
AFT-II	390.55	~2000	459.36	30.62	62.5	~300	5800	
BcIII	~300	1449.17	1458.42	820.84	307	~900	5700	
CGTX-II	165	>1000	>1000	>1000	105	133		
BgII		300		1000	500			
BgIII		18,900		9800	5100			
δ-AITX-Bcg1a	453				440	740		

## Data Availability

Not applicable.

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
