# Peer review of "The Sea Anemone Neurotoxins Modulating Sodium Channels: An Insight at Structure and Functional Activity after Four Decades of Investigation"

_toxins, 2022, doi:10.3390/toxins15010008_

Round 1
Reviewer 1 Report
This is an interesting review on sea anemone toxins, which summarizes structure/function relationships and recent evolutionary findings. Besides a thorough revision of the English language I have only few points as listed below:
p. 1, end of l. 32: and medicine
p. 1, l. 34: problems of
p. 1: sea anemones are not the oldest predatory organisms. Many protists have a predatory life style as well.
p. 1, l. 36: more than
p. 1, l. 37-39: rephrase sentence. Too long and a word is missing after popular
p. 1, l. 40: the ground
p. 2, l. 45: I wouldn’t call peptide toxins “chemical” weapons, rather biochemical
p. 2, l. 53: suggest to add more recent references on nematocysts
p. 2, l. 54: might occur additionally
p. 2, l. 61: in the body column
p. 2, l. 62-63: give references
p. 2, l. 66-67: protein family
p. 2, l. 67: at the protein level
p. 2, l. 71: the given references do not address the anatomical structure of cnidocysts or the distribution of cnidocyst types along the body. This is analyzed in detail by a recent paper from the Technau lab and in Zenkert et al., 2011
p. 2, l. 91: delete “ones”
p. 3, l. 104: were shown
p. 3, l. 124: The beta-subunit, according to electron
p. 3, l. 140: are expressed
p. 4, l. 158: causing
p. 4, l. 194: opening parentheses missing
p. 6, l. 232: the majority of the studied sea anemone toxins to date
p. 10, l. 403: secondary
p. 11, l. 419: high conservation
p. 11, l. 422: alpha-helix and beta-sheet
Fig. 10: using colors would make this figure much more readable. Also label termini with N and C
p. 12, l. 433: in Figure 10
p. 12, l. 442: delete “a”
p. 12, l. 455: 48aa is the length, not amino acid composition
p. 12, l. 455-456: which types are meant here? Specify
p. 12, l. 458: which type of two residues are meant here?
p. 12, l. 458-459 the N-terminus..the C-terminus
p. 12, l. 458-469: what could be the function of charged residues at the protein’s termini? Membrane interaction?
p. 12, l. 470-479: what is the evidence for aromatic residues having a role in protein-protein contacts for this system? In small cysteine-rich proteins I would rather expect a structural function in inducing turns to accommodate for the tight cysteine linkage pattern (in particular Trp).
p. 16, l. 631: what does Kb stand for? Is that not Kd?
p. 21, l. 822: the term “double-stranded peptide” is somewhat misleading, better use disulfide-linked dimer
Author Response
Dear reviewer, we are sincerely grateful to you for the great work on reviewing this manuscript, for your editing, valuable comments and suggestions aimed at improving the quality of the manuscript.
We have made changes and editing the text of the manuscript according to your questions and editing:
- 1, end of l. 32: and medicine.
Answer:
corrected, inserted “and”.
- 1, l. 34: problems of
Answer:
corrected, inserted “of”.
- 1: sea anemones are not the oldest predatory organisms. Many protists have a predatory life style as well.
Answer:
Thank you for your comment and clarification, with which I fully agree. This sentence has been replaced by the following: «Sea anemones, one of the most ancient and preserved predatory venomous animals, are widespread at all latitudes of the ocean».
- 1, l. 36: more than
Answer:
corrected, inserted word “then”.
- 1, l. 37-39: rephrase sentence. Too long and a word is missing after popular
Answer:
This sentence has been replaced by the:
“They have established themselves in venomics as one of the popular sources of proteinaceous and peptide toxins. These toxins belong to different structural types and interact with a variety of biological targets. Many of them have promising pharmacological activity”.
- 1, l. 40: the ground
Answer:
corrected, inserted “the” before ground.
- 2, l. 45: I wouldn’t call peptide toxins “chemical” weapons, rather biochemical
Answer:
Thank you for your comment, you are right, and we agree, most likely it is a biochemical weapon. We corrected this phrase on “remarkable cellular and biochemical weapon” and added new reference [10] to once [3], which was inserted in the References section instead of the old reference ([10] Fautin, D.F.; Mariscal, R.N. Cnidaria: Anthozoa. In Microscopic anatomy of invertebrates; Harrison, F.G., Westfall, J.A. Eds.; Wiley-Liss, Inc. : New York, 1991; Volume 2, pp. 267–358).
- 2, l. 53: suggest to add more recent references on nematocysts
Answer:
- More recent references on nematocysts:
[10] Karabulut, A.; McClain, M.; Rubinstein, B.; Sabin, K.Z.; McKinney, S.A.; Gibson, M.C. The architecture and operating mechanism of a cnidarian stinging organelle. Nat. Commun. 2022, 13, 3494. doi: 10.1038/s41467-022-31090-0.
and
[12] Zenkert, C.; Takahashi, T.; Diesner, M.-O.; Özbek, S. Morphological and Molecular Analysis of the Nematostella vectensis Cnidom. PLoS ONE 2011, 6, e22725. doi: 10.1371/journal.pone.0022725) are included in the text instead of older ones.
- 2, l. 54: might occur additionally
Answer:
corrected, inserted word “additionally”.
- 2, l. 61: in the body column
Answer:
corrected, inserted phrase “in the body”.
- 2, l. 62-63: give references
Answer:
corrected, inserted reference [17]
- 2, l. 66-67: protein family
Answer:
Thanks for the edit, proteins changed to “protein”
- 2, l. 67: at the protein level
Answer:
Thanks for the edit, indefinite article changed to definite
- 2, l. 71: the given references do not address the anatomical structure of cnidocysts or the distribution of cnidocyst types along the body. This is analyzed in detail by a recent paper from the Technau lab and in Zenkert et al., 2011.
Answer:
Thanks for the fair comment. References [5,8] removed, updated references [10,12] inserted instead:
[10] Karabulut, A.; McClain, M.; Rubinstein, B.; Sabin, K.Z.; McKinney, S.A.; Gibson, M.C. The architecture and operating mechanism of a cnidarian stinging organelle. Nat. Commun. 2022. 13(1), 3494. doi: 10.1038/s41467-022-31090-0. doi.org/10.1038/s41467-022-31090-0
[12] Zenkert, C.; Takahashi, T.; Diesner, M.-O.; Özbek, S. Morphological and Molecular Analysis of the Nematostella vectensis Cnidom. PLoS ONE 2011, 6, e22725. doi: 10.1371/journal.pone.0022725.
- 2, l. 91: delete “ones”
Answer:
corrected, removed "ones"
- 3, l. 104: were shown
Answer:
corrected, “wos” changed to “were”
- 3, l. 124: The beta-subunit, according to electron
Answer:
corrected, we added the definite article “the” to β-subunit and removed the definite article the from electron .....
- 3, l. 140: are expressed
Answer:
corrected, we have changed done to “expressed”
- 4, l. 158: causing
Answer:
corrected, we have changed causes to causing
- 4, l. 194: opening parentheses missing
Answer:
corrected, opening parenthese inserted
- 6, l. 232: the majority of the studied sea anemone toxins to date
Answer:
corrected, we changed the phrase "The majority studied to date sea anemone neurotoxins…" to "The majority of the studied sea anemone neurotoxins to date…"
- 10, l. 403: secondary
Answer:
Thank you, of course, “secondary”, not second.
corrected
- 11, l. 419: high conservation
Answer:
corrected, the word “conservativeness” has been replaced with “conservation”
- 11, l. 422: alpha-helix and beta-sheet
Answer:
Thanks! We have clarified and included the full names of the secondary structure elements: the words –“helix and –sheets”.
Fig. 10: using colors would make this figure much more readable. Also label termini with N and C
Answer:
corrected this figure, we inserted “N” and “C“ termini into all neurotoxins
- 12, l. 433: in Figure 10
Answer:
corrected, we have replaced the preposition on with “in”.
- 12, l. 442: delete “a”
Answer:
corrected, we removed the indefinite article.
- 12, l. 455: 48aa is the length, not amino acid composition
Answer:
corrected, we changed the phrase "aa in the amino acid composition" to "amino acid residues".
- 12, l. 455-456: which types are meant here? Specify
Answer:
We have replaced the sentence "The high degree (up to 98%) of structural homology is observed within one type of neurotoxins of different types" with "Within each of the four types of neurotoxins (1-4) there is a high degree (up to 98%) of structural homology".
- 12, l. 458: which type of two residues are meant here?
Answer:
We have refined this sentence by adding the phrase "mainly hydrophobic" to the word "residues": "A characteristic of the type 2 neurotoxins is the presence of only two mainly hyrophobic residues at the N-terminus of a sequence and of three-four basic residues at the C-terminus ....".
- 12, l. 458-459 the N-terminus the C-terminus
Answer:
corrected, we have inserted definite articles “the” to both terminus.
- 12, l. 458-469: what could be the function of charged residues at the protein’s termini? Membrane interaction?
Answer:
Yes, protein-protein interaction with membrane Navs and adjacent phospholipids
Previously, according to Kalina et al., 2020 [168], devoted to the study of Radianthus neurotoxins, in particular, in silico binding of δ-SHTX-Hcr1f (one from six representatives of Radianthus neurotoxins famile) with potential-sensing domains (VSD-I, VSD-IV) and pore-forming domains (PD1) of rNaV1.2, such charged residues of neurotoxin as Lys46, Lys48, Glu31, Asp7, and Arg45 interact with the Asp317, Lys355, Gly318, Asn340, and Asn285 of the extracellular region of the PDI. At the same time, the charged residues Asp6, Arg13, and Lys32 participate in interaction with the extracellular region of VSD-IV of NaV1.2 which includes besides Val1620 also two charged residues, Glu1616 and Lys1617 ([168] Figure 8). According to the calculated data, the charged C-terminal residues (Lys48, Lys46, Glu31) and Asp7 coordinate the toxin’s orientation at the extracellular region of the rNaV1.2 pore domain through a network of electrostatic interactions and hydrogen bonds. These residues are assumed to be mainly responsible for the binding of δ-SHTX-Hcr1f to this channel subtype, since they make a maximum contribution ( - 71.39, - 42.289, - 26.446, and - 23.855 kcal/mol, respectively) to binding energy. However, the presence of a protein-lipid interaction of Arg13 residues of δ-SHTX-Hcr1f with the membrane adjacent phospholipids should also be noted.
- 12, l. 470-479: what is the evidence for aromatic residues having a role in protein-protein contacts for this system? In small cysteine-rich proteins I would rather expect a structural function in inducing turns to accommodate for the tight cysteine linkage pattern (in particular Trp).
Answer:
Indeed, in silico calculated data for the complex of the neurotoxin δ-SHTX-Hcr1f with rNaV1.2 channel published earlier ([168], Figure 8c, Supplementary material) indicate that, in addition to protein-protein electrostatic and hydrophobic interactions, the formation of intermolecular π-π interactions between the aromatic residues of neurotoxin and aromatic residues of rNav1.2 (between Trp30 and Trp34 of δ-SHTX-Hcr1f and Trp302 of rNaV1.2) cause greater affinity of neurotoxin to channel. π-π Interactions introduce additionally energy to the formation of the complex and also play a stabilizing role for complex. This corresponds to the literature data [81].
- 16, l. 631: what does Kb stand for? Is that not Kd?
Answer:
Lines 631. We had in mind the equilibrium dissociation constant of the resulting complex of neurotoxins with Navs (KD). This table and table caption has been corrected, we inserted the sentence “The equilibrium dissociation constant of the resulting complex of neurotoxins with rat brain synaptosomal Navs”).
- 12, l. 470-479: what is the evidence for aromatic residues having a role in protein-protein contacts for this system? In small cysteine-rich proteins I would rather expect a structural function in inducing turns to accommodate for the tight cysteine linkage pattern (in particular Trp).
Answer:
Indeed, in silico calculated data published earlier ([168], Figure 8c, Supplementary material) for the complex of the neurotoxin δ-SHTX-Hcr1f with rNaV1.2 indicates the formation of intermolecular π-π interactions between the aromatic residues of a neurotoxin (Trp30 and Trp24)and aromatic residues of rNav1.2 (Trp303 and Trp1565]. The aromatic residues of neurotoxin cause their greater affinity to their targets. Several π-π interactions introduce additional energy into protein-protein electrostatic and hydrophobic interactions and also play a stabilizing role in the formation of the complex, which is consistent with the literature data [81].
- 21, l. 822: the term “double-stranded peptide” is somewhat misleading, better use disulfide-linked dimer.
Answer:
We named the peptide RTX-VI “double-stranded peptide” because its amino acid sequence completely repeated the one of the most toxic neurotoxin of the Radianthus family, RTX-III, except the amino acid residue Arg13. This residue was apparently lost during the post-translational modification of RTX-III. The amino acid sequence of the new RTX-VI peptide thus consists of two chains linked by two disulfide bridges (between CI-CV and CII-CIV), which are similar to those of the RTX-III neurotoxin. As it was shown earlier this process of the residues loss during post-translational modification occurs in so-called double-stranded spider toxins [Langenegger, N.; Koua, D.; Schürch, S.; Heller, M.; Nentwig, W.; Kuhn-Nentwig, L. Identification of a precursor processing protease from the spider Cupiennius salei essential for venom neurotoxin maturation. J. Biol. Chem. 2018, 293, 2079–2090.].
In addition, we provide line-by-line editing of all the corrections made in the latest version:
Line 12 – 20: We inserted three sentence “Therefore, a large number of publications are currently dedicated to the search and study of the structure-functional relationships of new sea anemone natural neurotoxins – potential pharmacologically active compounds that specifically interact with various subtypes of voltage gated sodium channels as drug discovery targets. This review presents and summarizes some updated data on the structure-functional relationships of known sea anemone neurotoxins belonging to four structural types. The review also emphasizes the study of type 2 neurotoxins, produced by the tropical sea anemone Heteractis crispa, five structurally homologous and one unique double-stranded peptide that, due to the absence of a functionally significant Arg14 residue, loses toxicity but retains the ability to modulate several VGSCs subtypes” instead old sentence “Therefore, searching and studying new highly selective neurotoxins interacting specifically and ef-12 fectively with various NaV subtypes, the creation of pharmacologically active molecules on their 13 basis for medicine, as well as the development of effective bioinsecticides for reducing losses in 14 agriculture are the actual direction of modern scientific researches. This review summarizes the in-15 formation and provides the updated recent data on the structure and the structure-functional rela-16 tionships of known sea anemone neurotoxins, most of which belong to four structural types. The 17 review also emphasizes the study of type 2 neurotoxins produced by the tropical sea anemone Het-18 eractis crispa, one of which, due to the absence of a functionally significant Arg14 residue, is a unique 19 double-stranded peptide having lost toxicity but interacting with a number of VGSCs subtypes.
Line 26: word “are” was changed to “is”.
Line 27: “ones“was changed to “ones“.
Line 33: we added “and” before medicine.
Line 33-34: a sentence has been inserted “Sea anemones, one of the most ancient and preserved predatory venomous animals, are widespread at all latitudes of the ocean” instead “Sea anemones, the oldest predatory or-34 ganisms, are widespread in all latitudes of the World Ocean”.
Line 40: the definite article "the" is inserted before “ground”.
Line 42: it is inserted “their” before “algal”.
Line 43: it is inserted the word "Moreover" instead of "besides".
Line 45: phrase “remarkable cellular and biochemical” is inserted instead “chemical weapon" and “as” instead “because”.
Line 47: it is inserted the word “Therefore” instead “Thus”.
Line 54: it is inserted the word “additionally”.
Line 57: it is inserted the word “body” before “column”.
Line 61: it is inserted the word “in the body” before “column” as well “the body” before next word “column”.
Line 64: a reference [17] is inserted at the end of the sentence and “the” перед “sea anemone life cycle”.
Lines 64, 67, 72: definite article “the” is inserted.
Line 66: word “proteins” changed to “protein”.
Line 71: references [5,8] we changed to [10].
Line 74: “takes place” instead “occurs”.
Line 75: “vary” instead “varies”.
Line 79: it is removed "and" before "now covers”.
Line 84: it is inserted the word “about” instead “near”.
Line 85: it is inserted “by” instead “with”.
Line 88: it is added “ly” to word “unprecedented”.
Line 89: it is added “ing” to “belong”.
Line 91: comma after word “known” is removed.
Line 97: “from” is changed to “of”.
Line 104: “was” is changed to “were”.
Line 108: indefinite article “a” before “NaVs functional activity” is removed.
Line 113: “and also” is replaced by “as well as”.
Line 124: “β-Subunit” is replaced by “The β-subunit”.
Line 133: indefinite article “a” before “cell” is removed.
Line 140: “done” is changed to “expressed”
Line 144: word “case” instead “occurrence”.
Line 148: “so-called” instead “so named”.
Line 150: “in” instead “to”.
Lines 151, 152, 153, 162: it is inserted article “the”
Line 153: “Thus” instead “so”.
Line 154: It is repositioned on the line “fast NaV inactivation”.
Line 158: “causing” instead “causes”.
Line 193: “prey” instead “preys”.
Lines 221, 253, 279: The sentence “The disulfide bridges are shown above the alignment” is inserted.
Line 231: “the arsenal of neurotoxins” instead “a neurotoxins arsenal”
Line 231: “The majority of the studied sea anemone neurotoxins to date” instead “The majority studied to date sea anemone neurotoxins”.
Line 257: “been” after “have” is deleted.
Line 259, 264: “so-called” instead “so named”.
Line 262: “to” instead “with”.
Line 268: “Figure 3a” instead “Figure 3”, “Thus” instead “So”.
Line 270: “others longer neurotoxins” instead “other more long neurotoxins”.
Line 275: Figure 3 is changed (Figures 3,4,5 is combined instead).
Line 279: added “(b) type 4 NaTxs CLX-1 (UniProt ID: P14531) and CLX-2 (P49127) from C. parasitica [132]; (c) Halcurin (UniProtKB: P0C5G6) from H. carlgreni [135] and Nv1 (B1NWS1) from N. vectensis [137]. The sequences of two Nv1 isoforms with С-terminal Q (B1NWS1-1) and K (B1NWS1-2) are shown [https://www.uniprot.org/uniprot/B1NWS1] [138] C-terminal residues of both isoforms are underlined.”
Line 282, 284: indefinite article “a” is changed on “the”.
Line 288: it is inserted “that of” before “crustaceans”.
Line 294: (Figure 3b) instead (Figure 4) .
Line 296: Figure 4 is deleted.
Line 303: “whose” instead “which”.
Line 304: “those” instead “ones”.
Line 311: “Figure 3c “ instead “Figure 5”.
Line 312: Figure 5 is deleted.
Line 317: “in a way that this” instead “so this”.
Line 319: “rNaV1.4-β-1, hNaV1.5-β-1” instead “rNaV1.4-beta-1/SCN4A-319 SCN1B, hNaV1.5-beta-1/SCN5A-beta-1”.
Line 320: “the rNav1.2a-β-1” instead “rNav1.2a-beta-320 1/SCN2A-SCN1B”.
Line 322: “et al.” instead “et co-workers”.
Line 323: “Moreover” instead “Besides”.
Line 329: “is” instead “was”.
Line 334: “Figure 4” instead “Figure 6”.
Line 344: “have been” instead “were”.
Line 346: caption to Figure 4 “ Figure 4” instead “Figure 6” as well“Identical amino acid residues are shown on a gray background” is inserted.
Line 348: “from” instead “in”.
Lines 351, 354: “Figure 5a” instead “Figure 7”.
Line 353: ”Moreover» instead “Besides “.
Line 355: Instead of Figure 7, a combined Figure 5 (from the old Figures 7, 8, and 9) is inserted.
Line 356: caption to Figure 5 “Figure 5” instead “Figure 7”.
Line 357: it is inserted “(b) BDS-I and CGTX-II from B. cangicum [106]; (c) structurally new class of so-called BcIV- and APETx-like peptides [107]: BcIV (UniProt ID: P84919) from B. caissarum [141], APETx1 (P61541) [41] and APETx2 (P61542) [42] from A. elegantissima; Am-II (P69930) from A. maculata [97]; BDS-I and BDS-II (P59084) [142] from A. sulcata. Hydroxy-Pro at position 24 of Am-II is denoted by ‘O’. Identical and conserved amino acid residues are shown on a dark and light gray background, respectively; the multiple sequence alignment was performed using Vector NTI software”.
Line 359: “However” instead “But”.
Line 362: “Figures 5” instead “Figures 7, 8”
Line 364: it is inserted (Figure 5b).
Lines 365, 366: it is deleted Figure 8.
Line 371: “Figure 5c” instead Figure 9.
Line 372: it is deleted Figure 9.
Line 382: “Figure 5c” instead “Figure 9”.
Line 384: “(Figures 5a-c) is inserted.
Line 389: “hERG” instead “HERG”.
Line 395: “Figure 3a” instead “Figure 3“, “However” instead “But”, inserted “which” before “highly”.
Line 397: “and” instead “as well as”.
Line 398: “Therefore” instead “Thus”.
Line 400: “the Avi 9a-1 and Avi 9a-2 toxins” instead “the toxins Avi 9a-1 and Avi 9a-2”.
Line 403: ”Secondary” instead “Second”.
Line 406: “adopted” instead “adopt”.
Line 410: “so-called” instead “so named”.
Line 411: “Figures 6a-f, and 6h, i” instead “Figure 10a-f, and 10h,i”.
Line 415: “permit to” instead “assume”.
Line 417: “yet been determined experimentally” instead “experimentally been determined”.
Line 419: “conservation” instead “conservativeness”.
Line 422: “alpha-helices and beta-sheets” instead “(alpha and beta)”.
Line 424: “Figure 6g” instead “Figure 10g”.
Line 425: “Figure 6” instead “Figure 10”.
Line 425: it is added the letters N and C to the corresponding N- and C-terminal residues of all neurotoxins in Figures 6.
Line 433: “in the Figure 6” instead “on the Figure 10”.
Line 434: “a magnitude and a direction”: it is removed indefinite articles.
Line 441: “1960s” instead “60s of the last century”.
Line 455: “ amino acid residues” instead “aa in the amino acid composition”.
Line 456: sentence “Within each of the four types of neurotoxins (1-4), there is a high (up to 98%) degree of structural homology” instead “The high degree (up to 98%) of structural homology is observed within one 456 type of neurotoxins of different types”.
Line 458: “mainly hydrophobic residues at the” instead “residues at N-458 terminus”.
Line 459: “at the C-terminus which” instead “at C-terminus what”.
Line 464: “of them” instead “ones”.
Line 474: “have none” instead “do not have or have only”.
Line 487: “of the peptides dipole moment which” instead “of a peptides dipole moment what”.
Line 493: “ApA was” instead “ApA is”.
Line 494: “ApB had” instead “ApB has”.
Line 495: “differed” instead “differ”, “showed” instead “show”.
Line 497: “were powerful” instead “are powerful”.
Line 498: “NaTxs acted” instead “NaTxs act”.
Line 502: “it differed” instead “it differs”.
Line 506: it is deleted “as”.
Line 520: “However” instead “But”.
Line 527: “showed that” instead “shows that”.
Line 530: “they provided” instead “they provide”.
Line 534: “ApB interacted” instead “ApB interacts”.
Line 545: “Therefore” instead “thus”.
Line 547: “were characterized by” instead“ are characterized by”.
Line 549: “these neurotoxins had” instead “these neurotoxins have”, “occupied only” instead “occupies”.
Line 568: “the molecular surface” instead “a molecular surface”, “Thus” instead “So”.
Line 591: “located in” instead “locating”.
Line 602: “Figures 1,2,5,6” instead “Figure 9”.
Lines 614, 622, 624: it is inserted definite article “the”.
Line 627: “than that” instead “than one”.
Line 630: Table 1 is formatted to better separate rows with neurotoxins, “Kb” in the “Binding to synaptosomes” column changed to “KD”.
Line 631: “The equilibrium dissociation constant of the resulting complex of neurotoxins with rat brain synaptosomal Navs” instead “Constant of neurotoxin binding to rat brain 631 synaptosomes.”
Line 635: “compared” instead “comparing”.
Line 640: “appears to be” instead “appears”.
Line 643: “Thus,” instead “So”.
Line 647: “the RTX-III action on the” instead “an RTX-III action on a”.
Line 650: “being practically” instead “are practically”.
Line 654: “type 2 representatives” instead “representatives type 2”.
Line 668: “equilibrium dissociation constant of the complex (KD)” instead “the constants of neurotoxin binding (Kb)”.
Line 681: “These sea anemone” instead “This”.
Line 682: “whose genes” instead “which genes”.
Line 685: “could be” instead “can be”.
Line 689: “Moran et al.” instead “Moran et co-workers”.
Line 693: “each of the species which” instead “each species what”.
Line 696: “more highly” instead “several highly”.
Line 698: “the progress” instead “a progress”.
Line 702: “1970s” instead “70s of the last century.
Line 721: Table 2 is formatted to better separate rows.
Line 724: “Mahnir et al.” instead “Mahnir et co-workers”.
Line 727: “showed” instead “have shown”.
Line 728: “neurotoxins was” instead “neurotoxins is”.
Line 730: “were distinguished” instead “are distinguished”.
Line 731: “showed that an ‘Arg loop’ ” instead “shown that a ‘Arg loop’ ”.
Line 735: “Thus,” instead “So”.
Line 740: “This indicated” instead “This indicates”, “were important” instead “are important”.
Line 745: “namely” instead “named”.
Lines 752, 754, 758, 766: “Figure 7” instead “Figure 11”.
Line 755: “The disulfide bridges are shown above the alignment. Identical residues are shown in grey” is inserted.
Line 765: “satisfied the β-defensin-like” instead “satisfies a β-defensin-like”.
Line 768: “emergence” instead “appearance”.
Line 780: “have been” instead “were”.
Line 788: “affect” instead “effect”.
Line 791: “unlike the type 1” instead “unlike type 1”.
Line 797, 806: “Kalina et al.” instead “Kalina et co-workers”.
Line 806: “Olivera et al.” instead “Olivera et coworkers”.
Line 812: “NaV1.1–NaV1.7” instead “NaV1.1–1.7”.
Line 813: “Thus,” instead “So”.
Line 817: “which is stipulated” instead “what is stipulated”.
Line 819: “NaTxs” instead “NaTx”.
Lines 827, 829: “the interaction” instead “an interaction”.
Line 834: “the neurotoxin activity” instead “a neurotoxin activity”.
Line 836: “the channel binding” instead “a channel binding”.
Line 838: “the VSD-IV domain” instead “a VSD-IV domain”.
Line 841: “the pore-forming” instead “a pore-forming”.
Line 842: “whose Asp317” instead “which Asp317”.
Line 848: “Thus,” instead “So”.
Line 851: “S3–S4 loop hindered” instead “S3–S4 loop hinders”.
Line 852: “the S4 helix” instead “a S4 helix”, “which prevented” instead “what prevents”.
Line 854: “revealed the set” instead “reveals the set”.
Line 858: “insect NaV channels, comparing to the mammalian ones” instead “mammalian NaV channels, comparing to insect ones”.
Line 861: “Figure 8” instead “Figure 12”.
Line 862: “i.e.,” instead “i.e.”.
Line 866: “components of currents and slow down” instead “components of currents; they also slow down”.
Line 868: “Figure 8” instead “Figure 12”.
Line 867: Changes have been made to Figure 8.
Line 889: “residue, which results” instead “residue what results”.
Line 890: “of electrophysiological” instead “on electrophysiological”.
Line 895: “The sea anemone” instead “Sea anemone”.
Line 897: “NaV subtypes as well as” instead “NaV subtypes, as well as”.
Line 899: “of further design” instead “of a further design”.
Line 919: reference “Karabulut, A.; McClain, M.; Rubinstein, B.; Sabin, K.Z.; McKinney, S.A.; Gibson, M.C. The architecture and operating mechanism of a cnidarian stinging organelle. Nat. Commun. 2022, 13, 3494. doi: 10.1038/s41467-022-31090-0” instead “Fautin, D.F.; Mariscal, R.N. Cnidaria: Anthozoa. In Microscopic anatomy of invertebrates; Harrison, F.G., Westfall, J.A. Eds.; 919 Wiley-Liss, Inc: New York, 1991; Volume 2, pp. 267–358”.
Line 923: reference “Zenkert, C.; Takahashi, T.; Diesner, M.-O.; Özbek, S. Morphological and Molecular Analysis of the Nematostella vectensis Cnidom. PLoS ONE 2011, 6, e22725. doi: 10.1371/journal.pone.0022725.” instead “Fautin, D.G. Structural diversity, systematics, and evolution of cnidae. Toxicon 2009, 54, 1054–1064. doi: 923 10.1016/j.toxicon.2009.02.024.”
Lines 952, 971, 978, 1017, 1020, 1023, 1036, 1046, 1056, 1062, 1087, 1113, 1122, 1144, 1147, 1198, 1223, 1253, 1283, 1306, 1321, 1332, 1347: blue font color formed in the doi and/or surnames of the authors in the references of the original version of the manuscript [24, 32, 35, 50, 51, 52, 57, 61, 66, 69, 80, 90, 95, 104, 105, 121, 126, 137, 150, 163, 174, 180, 185, 192], we changed to black color.
Reviewer 2 Report
This manuscript reviews the current literature investigating sodium channel-modulating toxins and their molecular targets, as well as their potential applied usage. this is an exciting topic and clearly, the authors have reviewed the literature extensively. However, the review is full of grammatical errors and even sometimes wrong information. Below are just some examples from the abstract and introduction.
The sentence in the abstract on line 12 is very long and hard to follow. It also ends with “are the actual direction of modern scientific researches” doesn’t make sense
The sentence on line 15 also doesn’t read so well
Line 17 I think you mean sea anemone neurotoxins modulating sodium channels.
Sentence line 33 doesn’t read well I think the authors meant “problems in these fundamental and applied sciences…”
Also, this sentence lacks references
I don’t think sea anemones can be considered the oldest predatory organism. I'm not sure what is meant by the oldest organism, maybe animal lineage is better. But also, all cnidarians are predatory as well. I think I understand what it meant but what is stated is incorrect. What is meant is cnidarians are likely the oldest extant venomous lineage.
This makes reviewing this manuscript extremely difficult. At its current stage, this needs to first be improved before the science can be adequately reviewed. I would also strongly suggest combing some of the figures to make composite figures. For example figure 3 4 and 5 can easily be combined.
Author Response
Dear reviewer, we are sincerely grateful to you for the great work on reviewing this manuscript, for your editing, valuable comments and suggestions aimed at improving the quality of the manuscript.
Below are the answers to your questions and the edit you suggested.
However, the review is full of grammatical errors and even sometimes wrong information. Below are just some examples from the abstract and introduction.
Answer:
We have conducted an extensive review of the English language in conjunction with our native speaker, which is a professional English translator. All edits have been made to edited version and are reflected in the review version of the manuscript.
The sentence in the abstract on line 12 is very long and hard to follow. It also ends with “are the actual direction of modern scientific researches” doesn’t make sense
Answer:
The sentence on line 12: "Therefore, searching and studying new highly selective neurotoxins interacting specifically and effectively with various NaV subtypes, the creation of pharmacologically active molecules on their basis for medicine, as well as the development of effective bioinsecticides for reducing losses in agriculture are the actual direction of modern scientific researches"
we have changed to: "Therefore, a large number of publications are currently devoted to the search and study of the structure-functional relationships of new sea anemone natural neurotoxins – potential pharmacologically active compounds that specifically interact with various subtypes of voltage gated sodium channels as drug discovery targets".
The sentence on line 15 also doesn’t read so well.
Answer:
The poorly readable sentence on line 15: "This review summarizes the information and provides the updated recent data on the structure and the structure-functional relationships of known sea anemone neurotoxins, most of which belong to four structural types” we have changed to: “This review presents and summarizes some updated data on the structure-functional relationships of known sea anemone neurotoxins belong to four structural types”.
Line 17 I think you mean sea anemone neurotoxins modulating sodium channels.
Answer:
The sentence on line 17: "The review also emphasizes the study of type 2 neurotoxins produced by the tropical sea anemone Heteractis crispa, one of which, due to the absence of a functionally significant Arg14 residue, is a unique double-stranded peptide having lost toxicity but interacting with a number of VGSCs subtypes” we have changed to: “The review also emphasizes the study of type 2 neurotoxins, produced by the tropical sea anemone Heteractis crispa, five structurally homologous and one unique double-stranded peptide that, due to the absence of a functionally significant Arg14 residue, loses toxicity but retains the ability to modulate several VGSCs subtypes”.
Sentence line 33 doesn’t read well I think the authors meant “problems in these fundamental and applied sciences…”. Also, this sentence lacks references.
Answer:
- As for the poorly readable sentence on line 33 "In recent decades marine bioresources have been widely used to solve many problems these fundamental and applied sciences",
we removed it from the text, since, indeed, it requires a lot of references from completely different areas (bioorganic chemistry, chemistry of natural compounds, etc.).
I don’t think sea anemones can be considered the oldest predatory organism. I'm not sure what is meant by the oldest organism, maybe animal lineage is better. But also, all cnidarians are predatory as well. I think I understand what it meant but what is stated is incorrect. What is meant is cnidarians are likely the oldest extant venomous lineage.
Answer:
- Thank you for your comment and clarification, with which we fully agree. This sentence has been replaced by the following: “Sea anemones, one of the most ancient and preserved predatory venomous animals, are widespread at all latitudes of the ocean”.
This makes reviewing this manuscript extremely difficult. At its current stage, this needs to first be improved before the science can be adequately reviewed. I would also strongly suggest combing some of the figures to make composite figures. For example figure 3 4 and 5 can easily be combined.
Answer:
We apologize for the length and style of the manuscript, which do not allow the reviewer to evaluate it. We have tried to correct the mistakes made, including the grammar of errors.
Thanks for reviewer suggestion to facilitate the manuscript by combining figures 3,4,5!!!
We combined these figures and turned them into Figure 3a-c. We have also combined figures 7,8,9 and turned them into figure 5a-c. So figure 6 became figure 4, figure 10 became figure 6, figure 11 became figure 7, and figure 12 became figure 8.
In addition, we provide line-by-line editing of all the corrections made in the latest version:
Line 12 – 20: We inserted three sentence “Therefore, a large number of publications are currently dedicated to the search and study of the structure-functional relationships of new sea anemone natural neurotoxins – potential pharmacologically active compounds that specifically interact with various subtypes of voltage gated sodium channels as drug discovery targets. This review presents and summarizes some updated data on the structure-functional relationships of known sea anemone neurotoxins belonging to four structural types. The review also emphasizes the study of type 2 neurotoxins, produced by the tropical sea anemone Heteractis crispa, five structurally homologous and one unique double-stranded peptide that, due to the absence of a functionally significant Arg14 residue, loses toxicity but retains the ability to modulate several VGSCs subtypes”
instead old sentence:
“Therefore, searching and studying new highly selective neurotoxins interacting specifically and effectively with various NaV subtypes, the creation of pharmacologically active molecules on their basis for medicine, as well as the development of effective bioinsecticides for reducing losses in agriculture are the actual direction of modern scientific researches. This review summarizes the information and provides the updated recent data on the structure and the structure-functional relationships of known sea anemone neurotoxins, most of which belong to four structural types. The review also emphasizes the study of type 2 neurotoxins produced by the tropical sea anemone Heteractis crispa, one of which, due to the absence of a functionally significant Arg14 residue, is a unique double-stranded peptide having lost toxicity but interacting with a number of VGSCs subtypes.
Line 26: word “are” was changed to “is”.
Line 27: “ones“was changed to “ones“.
Line 33: we added “and” before medicine.
Line 33-34: a sentence has been inserted “Sea anemones, one of the most ancient and preserved predatory venomous animals, are widespread at all latitudes of the ocean” instead “Sea anemones, the oldest predatory organisms, are widespread in all latitudes of the World Ocean”.
Line 40: the definite article "t" is inserted before “ground”.
Line 42: it is inserted “” before “algal”.
Line 43: it is inserted the word "Moreover" instead of "besides".
Line 45: phrase “remarkable cellular and biochemical” is inserted instead “chemical weapon" and “as” instead “because”.
Line 47: it is inserted the word “Therefore” instead “Thus”.
Line 54: it is inserted the word “additionally”.
Line 57: it is inserted the word “body” before “column”.
Line 61: it is inserted the word “in the body” before “column” as well “the body” before next word “column”.
Line 64: a reference [17] is inserted at the end of the sentence and “the” перед “sea anemone life cycle”.
Lines 64, 67, 72: definite article “the” is inserted.
Line 66: word “proteins” changed to “protein”.
Line 71: references [5,8] we changed to [10].
Line 74: “takes place” instead “occurs”.
Line 75: “vary” instead “varies”.
Line 79: it is removed "and" before "now covers”.
Line 84: it is inserted the word “about” instead “near”.
Line 85: it is inserted “by” instead “with”.
Line 88: it is added “ly” to word “unprecedented”.
Line 89: it is added “ing” to “belong”.
Line 91: comma after word “known” is removed.
Line 97: “from” is changed to “of”.
Line 104: “was” is changed to “were”.
Line 108: indefinite article “a” before “NaVs functional activity” is removed.
Line 113: “and also” is replaced by “as well as”.
Line 124: “β-Subunit” is replaced by “The β-subunit”.
Line 133: indefinite article “a” before “cell” is removed.
Line 140: “done” is changed to “expressed”
Line 144: word “case” instead “occurrence”.
Line 148: “so-called” instead “so named”.
Line 150: “in” instead “to”.
Lines 151, 152, 153, 162: it is inserted article “the”
Line 153: “Thus” instead “so”.
Line 154: It is repositioned on the line “fast NaV inactivation”.
Line 158: “causing” instead “causes”.
Line 193: “prey” instead “preys”.
Lines 221, 253, 279: The sentence “The disulfide bridges are shown above the alignment” is inserted.
Line 231: “the arsenal of neurotoxins” instead “a neurotoxins arsenal”.
Line 231: “The majority of the studied sea anemone neurotoxins to date” instead “The majority studied to date sea anemone neurotoxins”.
Line 257: “been” after “have” is deleted.
Line 259, 264: “so-called” instead “so named”.
Line 262: “to” instead “with”.
Line 268: “Figure 3a” instead “Figure 3”, “Thus” instead “So”.
Line 270: “others longer neurotoxins” instead “other more long neurotoxins”.
Line 275: Figure 3 is changed (Figures 3,4,5 is combined instead).
Line 279: added “(b) type 4 NaTxs CLX-1 (UniProt ID: P14531) and CLX-2 (P49127) from C. parasitica [132]; (c) Halcurin (UniProtKB: P0C5G6) from H. carlgreni [135] and Nv1 (B1NWS1) from N. vectensis [137]. The sequences of two Nv1 isoforms with С-terminal Q (B1NWS1-1) and K (B1NWS1-2) are shown [https://www.uniprot.org/uniprot/B1NWS1] [138] C-terminal residues of both isoforms are underlined.”
Line 282, 284: indefinite article “a” is changed on “the”.
Line 288: it is inserted “that of” before “crustaceans”.
Line 294: (Figure 3b) instead (Figure 4) .
Line 296: Figure 4 is deleted.
Line 303: “whose” instead “which”.
Line 304: “those” instead “ones”.
Line 311: “Figure 3c “ instead “Figure 5”.
Line 312: Figure 5 is deleted.
Line 317: “in a way that this” instead “so this”.
Line 319: “rNaV1.4-β-1, hNaV1.5-β-1” instead “rNaV1.4-beta-1/SCN4A-319 SCN1B, hNaV1.5-beta-1/SCN5A-beta-1”.
Line 320: “the rNav1.2a-β-1” instead “rNav1.2a-beta-320 1/SCN2A-SCN1B”.
Line 322: “et al.” instead “et co-workers”.
Line 323: “Moreover” instead “Besides”.
Line 329: “is” instead “was”.
Line 334: “Figure 4” instead “Figure 6”.
Line 344: “have been” instead “were”.
Line 346: caption to Figure 4 “ Figure 4” instead “Figure 6” as well “Identical amino acid residues are shown on a gray background” is inserted.
Line 348: “from” instead “in”.
Lines 351, 354: “Figure 5a” instead “Figure 7”.
Line 353: ”Moreover” instead “Besides “.
Line 355: Instead of Figure 7, a combined Figure 5 (from the old Figures 7, 8, and 9) is inserted.
Line 356: caption to Figure 5 “Figure 5” instead “Figure 7”.
Line 357: it is inserted “(b) BDS-I and CGTX-II from B. cangicum [106]; (c) structurally new class of so-called BcIV- and APETx-like peptides [107]: BcIV (UniProt ID: P84919) from B. caissarum [141], APETx1 (P61541) [41] and APETx2 (P61542) [42] from A. elegantissima; Am-II (P69930) from A. maculata [97]; BDS-I and BDS-II (P59084) [142] from A. sulcata. Hydroxy-Pro at position 24 of Am-II is denoted by ‘O’. Identical and conserved amino acid residues are shown on a dark and light gray background, respectively; the multiple sequence alignment was performed using Vector NTI software”.
Line 359: “However” instead “But”.
Line 362: “Figures 5” instead “Figures 7, 8”
Line 364: it is inserted (Figure 5b).
Lines 365, 366: it is deleted Figure 8.
Line 371: “Figure 5c” instead Figure 9.
Line 372: it is deleted Figure 9.
Line 382: “Figure 5c” instead “Figure 9”.
Line 384: “(Figures 5a-c)” is inserted.
Line 389: “hERG” instead “HERG”.
Line 395: “Figure 3a” instead “Figure 3“, “However” instead “But”, inserted “which” before “highly”.
Line 397: “and” instead “as well as”.
Line 398: “Therefore” instead “Thus”.
Line 400: “the Avi 9a-1 and Avi 9a-2 toxins” instead “the toxins Avi 9a-1 and Avi 9a-2”.
Line 403: ”Secondary” instead “Second”.
Line 406: “adopted” instead “adopt”.
Line 410: “so-called” instead “so named”.
Line 411: “Figures 6a-f, and 6h, i” instead “Figure 10a-f, and 10h,i”.
Line 415: “permit to” instead “assume”.
Line 417: “yet been determined experimentally” instead “experimentally been determined”.
Line 419: “conservation” instead “conservativeness”.
Line 422: “alpha-helices and beta-sheets” instead “(alpha and beta)”.
Line 424: “Figure 6g” instead “Figure 10g”.
Line 425: “Figure 6” instead “Figure 10”.
Line 425: it is added the letters N and C to the corresponding N- and C-terminal residues of all neurotoxins in Figures 6.
Line 433: “in the Figure 6” instead “on the Figure 10”.
Line 434: “a magnitude and a direction”: it is removed indefinite articles.
Line 441: “1960s” instead “60s of the last century”.
Line 455: “amino acid residues” instead “aa in the amino acid composition”.
Line 456: sentence “Within each of the four types of neurotoxins (1-4), there is a high (up to 98%) degree of structural homology” instead “The high degree (up to 98%) of structural homology is observed within one 456 type of neurotoxins of different types”.
Line 458: “mainly hydrophobic residues at the” instead “residues at N-terminus”.
Line 459: “at the C-terminus which” instead “at C-terminus what”.
Line 464: “of them” instead “ones”.
Line 474: “have none” instead “do not have or have only”.
Line 487: “of the peptides dipole moment which” instead “of a peptides dipole moment what”.
Line 493: “ApA was” instead “ApA is”.
Line 494: “ApB had” instead “ApB has”.
Line 495: “differed” instead “differ”, “showed” instead “show”.
Line 497: “were powerful” instead “are powerful”.
Line 498: “NaTxs acted” instead “NaTxs act”.
Line 502: “it differed” instead “it differs”.
Line 506: it is deleted “as”.
Line 520: “However” instead “But”.
Line 527: “showed that” instead “shows that”.
Line 530: “they provided” instead “they provide”.
Line 534: “ApB interacted” instead “ApB interacts”.
Line 545: “Therefore” instead “thus”.
Line 547: “were characterized by” instead“ are characterized by”.
Line 549: “these neurotoxins had” instead “these neurotoxins have”, “occupied only” instead “occupies”.
Line 568: “the molecular surface” instead “a molecular surface”, “Thus” instead “So”.
Line 591: “located in” instead “locating”.
Line 602: “Figures 1,2,5,6” instead “Figure 9”.
Lines 614, 622, 624: it is inserted definite article “the”.
Line 627: “than that” instead “than one”.
Line 630: Table 1 is formatted to better separate rows with neurotoxins, “Kb” in the “Binding to synaptosomes” column changed to “KD”.
Line 631: “The equilibrium dissociation constant of the resulting complex of neurotoxins with rat brain synaptosomal Navs” instead “Constant of neurotoxin binding to rat brain synaptosomes.”
Line 635: “compared” instead “comparing”.
Line 640: “appears to be” instead “appears”.
Line 643: “Thus,” instead “So”.
Line 647: “the RTX-III action on the” instead “an RTX-III action on a”.
Line 650: “being practically” instead “are practically”.
Line 654: “type 2 representatives” instead “representatives type 2”.
Line 668: “equilibrium dissociation constant of the complex (KD)” instead “the constants of neurotoxin binding (Kb)”.
Line 681: “These sea anemone” instead “This”.
Line 682: “whose genes” instead “which genes”.
Line 685: “could be” instead “can be”.
Line 689: “Moran et al.” instead “Moran et co-workers”.
Line 693: “each of the species which” instead “each species what”.
Line 696: “more highly” instead “several highly”.
Line 698: “the progress” instead “a progress”.
Line 702: “1970s” instead “70s of the last century.
Line 721: Table 2 is formatted to better separate rows.
Line 724: “Mahnir et al.” instead “Mahnir et co-workers”.
Line 727: “showed” instead “have shown”.
Line 728: “neurotoxins was” instead “neurotoxins is”.
Line 730: “were distinguished” instead “are distinguished”.
Line 731: “showed that an ‘Arg loop’ ” instead “shown that a ‘Arg loop’ ”.
Line 735: “Thus,” instead “So”.
Line 740: “This indicated” instead “This indicates”, “were important” instead “are important”.
Line 745: “namely” instead “named”.
Lines 752, 754, 758, 766: “Figure 7” instead “Figure 11”.
Line 755: “The disulfide bridges are shown above the alignment. Identical residues are shown in grey” is inserted.
Line 765: “satisfied the β-defensin-like” instead “satisfies a β-defensin-like”.
Line 768: “emergence” instead “appearance”.
Line 780: “have been” instead “were”.
Line 788: “affect” instead “effect”.
Line 791: “unlike the type 1” instead “unlike type 1”.
Line 797, 806: “Kalina et al.” instead “Kalina et co-workers”.
Line 806: “Olivera et al.” instead “Olivera et coworkers”.
Line 812: “NaV1.1–NaV1.7” instead “NaV1.1–1.7”.
Line 813: “Thus,” instead “So”.
Line 817: “which is stipulated” instead “what is stipulated”.
Line 819: “NaTxs” instead “NaTx”.
Lines 827, 829: “the interaction” instead “an interaction”.
Line 834: “the neurotoxin activity” instead “a neurotoxin activity”.
Line 836: “the channel binding” instead “a channel binding”.
Line 838: “the VSD-IV domain” instead “a VSD-IV domain”.
Line 841: “the pore-forming” instead “a pore-forming”.
Line 842: “whose Asp317” instead “which Asp317”.
Line 848: “Thus,” instead “So”.
Line 851: “S3–S4 loop hindered” instead “S3–S4 loop hinders”.
Line 852: “the S4 helix” instead “a S4 helix”, “which prevented” instead “what prevents”.
Line 854: “revealed the set” instead “reveals the set”.
Line 858: “insect NaV channels, comparing to the mammalian ones” instead “mammalian NaV channels, comparing to insect ones”.
Line 861: “Figure 8” instead “Figure 12”.
Line 862: “i.e.,” instead “i.e.”.
Line 866: “components of currents and slow down” instead “components of currents; they also slow down”.
Line 868: “Figure 8” instead “Figure 12”.
Line 867: Changes have been made to Figure 8.
Line 889: “residue, which results” instead “residue what results”.
Line 890: “of electrophysiological” instead “on electrophysiological”.
Line 895: “The sea anemone” instead “Sea anemone”.
Line 897: “NaV subtypes as well as” instead “NaV subtypes, as well as”.
Line 899: “of further design” instead “of a further design”.
Line 919: reference “Karabulut, A.; McClain, M.; Rubinstein, B.; Sabin, K.Z.; McKinney, S.A.; Gibson, M.C. The architecture and operating mechanism of a cnidarian stinging organelle. Nat. Commun. 2022, 13, 3494. doi: 10.1038/s41467-022-31090-0” instead “Fautin, D.F.; Mariscal, R.N. Cnidaria: Anthozoa. In Microscopic anatomy of invertebrates; Harrison, F.G., Westfall, J.A. Eds.; 919 Wiley-Liss, Inc: New York, 1991; Volume 2, pp. 267–358”.
Line 923: reference “Zenkert, C.; Takahashi, T.; Diesner, M.-O.; Özbek, S. Morphological and Molecular Analysis of the Nematostella vectensis Cnidom. PLoS ONE 2011, 6, e22725. doi: 10.1371/journal.pone.0022725.” instead “Fautin, D.G. Structural diversity, systematics, and evolution of cnidae. Toxicon 2009, 54, 1054–1064. doi: 923 10.1016/j.toxicon.2009.02.024.”
Lines 952, 971, 978, 1017, 1020, 1023, 1036, 1046, 1056, 1062, 1087, 1113, 1122, 1144, 1147, 1198, 1223, 1253, 1283, 1306, 1321, 1332, 1347: blue font color formed in the doi and/or surnames of the authors in the references of the original version of the manuscript [24, 32, 35, 50, 51, 52, 57, 61, 66, 69, 80, 90, 95, 104, 105, 121, 126, 137, 150, 163, 174, 180, 185, 192], we changed to black color.

Reviewer 3 Report
This is a very comprehensive and nicely written review describing anemone toxins and their interaciton with sodium voltage gated channels.
I have only some concerns erlated to the use of term "inactivation" throughout the MS in relation to Nav channels. Usually, these channels can exist in one of three possible states: OPEN, CLOSED, INACTIVE. The term inactivation is usually reserved to the Open--->Inactive transition (ball-and-chain mechanism) whereas the term DEactivaton is used to the Open--->Closed transition (which depends on VSD conformational rearrangments in ersposne to changes in the elctric field). I believe the authors ment that toxins, by binding to the VSD, are affecting the DEactivation kinetics while they keep wrongly referring to inactivation.
Author Response
We appreciate for your attention and kindly spirit to our manuscript as well as for the valuable corrections and suggestions.
Please find below our answers on your comments:
In connection with your remark about the correct use of the inactivation and deactivation terms, we believe that in this manuscript it would be better to use the term inactivation. As for the H. crispa peptides, we did not perform additional experiments to find out how their effect on the channel affects the deactivation process. The literature, describing sea anemone neurotoxins, and our own observations suggest that the Heteractis neurotoxins inhibit fust inactivation: open to inactive state transition provided by ball-and-chain mechanism. Hence, according to your comment, the term inactivation would probably be more correct.
In addition, we provide line-by-line editing of all the corrections made in the latest version according to questions and editing of reviewers 1 and 2:
Line 12 – 20: We inserted three sentence “Therefore, a large number of publications are currently dedicated to the search and study of the structure-functional relationships of new sea anemone natural neurotoxins – potential pharmacologically active compounds that specifically interact with various subtypes of voltage gated sodium channels as drug discovery targets. This review presents and summarizes some updated data on the structure-functional relationships of known sea anemone neurotoxins belonging to four structural types. The review also emphasizes the study of type 2 neurotoxins, produced by the tropical sea anemone Heteractis crispa, five structurally homologous and one unique double-stranded peptide that, due to the absence of a functionally significant Arg14 residue, loses toxicity but retains the ability to modulate several VGSCs subtypes”
instead old sentence:
“Therefore, searching and studying new highly selective neurotoxins interacting specifically and effectively with various NaV subtypes, the creation of pharmacologically active molecules on their basis for medicine, as well as the development of effective bioinsecticides for reducing losses in agriculture are the actual direction of modern scientific researches. This review summarizes the information and provides the updated recent data on the structure and the structure-functional relationships of known sea anemone neurotoxins, most of which belong to four structural types. The review also emphasizes the study of type 2 neurotoxins produced by the tropical sea anemone Heteractis crispa, one of which, due to the absence of a functionally significant Arg14 residue, is a unique double-stranded peptide having lost toxicity but interacting with a number of VGSCs subtypes.
Line 26: word “are” was changed to “is”.
Line 27: “ones“was changed to “ones“.
Line 33: we added “and” before medicine.
Line 33-34: a sentence has been inserted “Sea anemones, one of the most ancient and preserved predatory venomous animals, are widespread at all latitudes of the ocean” instead “Sea anemones, the oldest predatory organisms, are widespread in all latitudes of the World Ocean”.
Line 40: the definite article "t" is inserted before “ground”.
Line 42: it is inserted “” before “algal”.
Line 43: it is inserted the word "Moreover" instead of "besides".
Line 45: phrase “remarkable cellular and biochemical” is inserted instead “chemical weapon" and “as” instead “because”.
Line 47: it is inserted the word “Therefore” instead “Thus”.
Line 54: it is inserted the word “additionally”.
Line 57: it is inserted the word “body” before “column”.
Line 61: it is inserted the word “in the body” before “column” as well “the body” before next word “column”.
Line 64: a reference [17] is inserted at the end of the sentence and “the” перед “sea anemone life cycle”.
Lines 64, 67, 72: definite article “the” is inserted.
Line 66: word “proteins” changed to “protein”.
Line 71: references [5,8] we changed to [10].
Line 74: “takes place” instead “occurs”.
Line 75: “vary” instead “varies”.
Line 79: it is removed "and" before "now covers”.
Line 84: it is inserted the word “about” instead “near”.
Line 85: it is inserted “by” instead “with”.
Line 88: it is added “ly” to word “unprecedented”.
Line 89: it is added “ing” to “belong”.
Line 91: comma after word “known” is removed.
Line 97: “from” is changed to “of”.
Line 104: “was” is changed to “were”.
Line 108: indefinite article “a” before “NaVs functional activity” is removed.
Line 113: “and also” is replaced by “as well as”.
Line 124: “β-Subunit” is replaced by “The β-subunit”.
Line 133: indefinite article “a” before “cell” is removed.
Line 140: “done” is changed to “expressed”
Line 144: word “case” instead “occurrence”.
Line 148: “so-called” instead “so named”.
Line 150: “in” instead “to”.
Lines 151, 152, 153, 162: it is inserted article “the”
Line 153: “Thus” instead “so”.
Line 154: It is repositioned on the line “fast NaV inactivation”.
Line 158: “causing” instead “causes”.
Line 193: “prey” instead “preys”.
Lines 221, 253, 279: The sentence “The disulfide bridges are shown above the alignment” is inserted.
Line 231: “the arsenal of neurotoxins” instead “a neurotoxins arsenal”.
Line 231: “The majority of the studied sea anemone neurotoxins to date” instead “The majority studied to date sea anemone neurotoxins”.
Line 257: “been” after “have” is deleted.
Line 259, 264: “so-called” instead “so named”.
Line 262: “to” instead “with”.
Line 268: “Figure 3a” instead “Figure 3”, “Thus” instead “So”.
Line 270: “others longer neurotoxins” instead “other more long neurotoxins”.
Line 275: Figure 3 is changed (Figures 3,4,5 is combined instead).
Line 279: added “(b) type 4 NaTxs CLX-1 (UniProt ID: P14531) and CLX-2 (P49127) from C. parasitica [132]; (c) Halcurin (UniProtKB: P0C5G6) from H. carlgreni [135] and Nv1 (B1NWS1) from N. vectensis [137]. The sequences of two Nv1 isoforms with С-terminal Q (B1NWS1-1) and K (B1NWS1-2) are shown [https://www.uniprot.org/uniprot/B1NWS1] [138] C-terminal residues of both isoforms are underlined.”
Line 282, 284: indefinite article “a” is changed on “the”.
Line 288: it is inserted “that of” before “crustaceans”.
Line 294: (Figure 3b) instead (Figure 4) .
Line 296: Figure 4 is deleted.
Line 303: “whose” instead “which”.
Line 304: “those” instead “ones”.
Line 311: “Figure 3c “ instead “Figure 5”.
Line 312: Figure 5 is deleted.
Line 317: “in a way that this” instead “so this”.
Line 319: “rNaV1.4-β-1, hNaV1.5-β-1” instead “rNaV1.4-beta-1/SCN4A-319 SCN1B, hNaV1.5-beta-1/SCN5A-beta-1”.
Line 320: “the rNav1.2a-β-1” instead “rNav1.2a-beta-320 1/SCN2A-SCN1B”.
Line 322: “et al.” instead “et co-workers”.
Line 323: “Moreover” instead “Besides”.
Line 329: “is” instead “was”.
Line 334: “Figure 4” instead “Figure 6”.
Line 344: “have been” instead “were”.
Line 346: caption to Figure 4 “ Figure 4” instead “Figure 6” as well “Identical amino acid residues are shown on a gray background” is inserted.
Line 348: “from” instead “in”.
Lines 351, 354: “Figure 5a” instead “Figure 7”.
Line 353: ”Moreover” instead “Besides “.
Line 355: Instead of Figure 7, a combined Figure 5 (from the old Figures 7, 8, and 9) is inserted.
Line 356: caption to Figure 5 “Figure 5” instead “Figure 7”.
Line 357: it is inserted “(b) BDS-I and CGTX-II from B. cangicum [106]; (c) structurally new class of so-called BcIV- and APETx-like peptides [107]: BcIV (UniProt ID: P84919) from B. caissarum [141], APETx1 (P61541) [41] and APETx2 (P61542) [42] from A. elegantissima; Am-II (P69930) from A. maculata [97]; BDS-I and BDS-II (P59084) [142] from A. sulcata. Hydroxy-Pro at position 24 of Am-II is denoted by ‘O’. Identical and conserved amino acid residues are shown on a dark and light gray background, respectively; the multiple sequence alignment was performed using Vector NTI software”.
Line 359: “However” instead “But”.
Line 362: “Figures 5” instead “Figures 7, 8”
Line 364: it is inserted (Figure 5b).
Lines 365, 366: it is deleted Figure 8.
Line 371: “Figure 5c” instead Figure 9.
Line 372: it is deleted Figure 9.
Line 382: “Figure 5c” instead “Figure 9”.
Line 384: “(Figures 5a-c)” is inserted.
Line 389: “hERG” instead “HERG”.
Line 395: “Figure 3a” instead “Figure 3“, “However” instead “But”, inserted “which” before “highly”.
Line 397: “and” instead “as well as”.
Line 398: “Therefore” instead “Thus”.
Line 400: “the Avi 9a-1 and Avi 9a-2 toxins” instead “the toxins Avi 9a-1 and Avi 9a-2”.
Line 403: ”Secondary” instead “Second”.
Line 406: “adopted” instead “adopt”.
Line 410: “so-called” instead “so named”.
Line 411: “Figures 6a-f, and 6h, i” instead “Figure 10a-f, and 10h,i”.
Line 415: “permit to” instead “assume”.
Line 417: “yet been determined experimentally” instead “experimentally been determined”.
Line 419: “conservation” instead “conservativeness”.
Line 422: “alpha-helices and beta-sheets” instead “(alpha and beta)”.
Line 424: “Figure 6g” instead “Figure 10g”.
Line 425: “Figure 6” instead “Figure 10”.
Line 425: it is added the letters N and C to the corresponding N- and C-terminal residues of all neurotoxins in Figures 6.
Line 433: “in the Figure 6” instead “on the Figure 10”.
Line 434: “a magnitude and a direction”: it is removed indefinite articles.
Line 441: “1960s” instead “60s of the last century”.
Line 455: “amino acid residues” instead “aa in the amino acid composition”.
Line 456: sentence “Within each of the four types of neurotoxins (1-4), there is a high (up to 98%) degree of structural homology” instead “The high degree (up to 98%) of structural homology is observed within one 456 type of neurotoxins of different types”.
Line 458: “mainly hydrophobic residues at the” instead “residues at N-terminus”.
Line 459: “at the C-terminus which” instead “at C-terminus what”.
Line 464: “of them” instead “ones”.
Line 474: “have none” instead “do not have or have only”.
Line 487: “of the peptides dipole moment which” instead “of a peptides dipole moment what”.
Line 493: “ApA was” instead “ApA is”.
Line 494: “ApB had” instead “ApB has”.
Line 495: “differed” instead “differ”, “showed” instead “show”.
Line 497: “were powerful” instead “are powerful”.
Line 498: “NaTxs acted” instead “NaTxs act”.
Line 502: “it differed” instead “it differs”.
Line 506: it is deleted “as”.
Line 520: “However” instead “But”.
Line 527: “showed that” instead “shows that”.
Line 530: “they provided” instead “they provide”.
Line 534: “ApB interacted” instead “ApB interacts”.
Line 545: “Therefore” instead “thus”.
Line 547: “were characterized by” instead“ are characterized by”.
Line 549: “these neurotoxins had” instead “these neurotoxins have”, “occupied only” instead “occupies”.
Line 568: “the molecular surface” instead “a molecular surface”, “Thus” instead “So”.
Line 591: “located in” instead “locating”.
Line 602: “Figures 1,2,5,6” instead “Figure 9”.
Lines 614, 622, 624: it is inserted definite article “the”.
Line 627: “than that” instead “than one”.
Line 630: Table 1 is formatted to better separate rows with neurotoxins, “Kb” in the “Binding to synaptosomes” column changed to “KD”.
Line 631: “The equilibrium dissociation constant of the resulting complex of neurotoxins with rat brain synaptosomal Navs” instead “Constant of neurotoxin binding to rat brain synaptosomes.”
Line 635: “compared” instead “comparing”.
Line 640: “appears to be” instead “appears”.
Line 643: “Thus,” instead “So”.
Line 647: “the RTX-III action on the” instead “an RTX-III action on a”.
Line 650: “being practically” instead “are practically”.
Line 654: “type 2 representatives” instead “representatives type 2”.
Line 668: “equilibrium dissociation constant of the complex (KD)” instead “the constants of neurotoxin binding (Kb)”.
Line 681: “These sea anemone” instead “This”.
Line 682: “whose genes” instead “which genes”.
Line 685: “could be” instead “can be”.
Line 689: “Moran et al.” instead “Moran et co-workers”.
Line 693: “each of the species which” instead “each species what”.
Line 696: “more highly” instead “several highly”.
Line 698: “the progress” instead “a progress”.
Line 702: “1970s” instead “70s of the last century.
Line 721: Table 2 is formatted to better separate rows.
Line 724: “Mahnir et al.” instead “Mahnir et co-workers”.
Line 727: “showed” instead “have shown”.
Line 728: “neurotoxins was” instead “neurotoxins is”.
Line 730: “were distinguished” instead “are distinguished”.
Line 731: “showed that an ‘Arg loop’ ” instead “shown that a ‘Arg loop’ ”.
Line 735: “Thus,” instead “So”.
Line 740: “This indicated” instead “This indicates”, “were important” instead “are important”.
Line 745: “namely” instead “named”.
Lines 752, 754, 758, 766: “Figure 7” instead “Figure 11”.
Line 755: “The disulfide bridges are shown above the alignment. Identical residues are shown in grey” is inserted.
Line 765: “satisfied the β-defensin-like” instead “satisfies a β-defensin-like”.
Line 768: “emergence” instead “appearance”.
Line 780: “have been” instead “were”.
Line 788: “affect” instead “effect”.
Line 791: “unlike the type 1” instead “unlike type 1”.
Line 797, 806: “Kalina et al.” instead “Kalina et co-workers”.
Line 806: “Olivera et al.” instead “Olivera et coworkers”.
Line 812: “NaV1.1–NaV1.7” instead “NaV1.1–1.7”.
Line 813: “Thus,” instead “So”.
Line 817: “which is stipulated” instead “what is stipulated”.
Line 819: “NaTxs” instead “NaTx”.
Lines 827, 829: “the interaction” instead “an interaction”.
Line 834: “the neurotoxin activity” instead “a neurotoxin activity”.
Line 836: “the channel binding” instead “a channel binding”.
Line 838: “the VSD-IV domain” instead “a VSD-IV domain”.
Line 841: “the pore-forming” instead “a pore-forming”.
Line 842: “whose Asp317” instead “which Asp317”.
Line 848: “Thus,” instead “So”.
Line 851: “S3–S4 loop hindered” instead “S3–S4 loop hinders”.
Line 852: “the S4 helix” instead “a S4 helix”, “which prevented” instead “what prevents”.
Line 854: “revealed the set” instead “reveals the set”.
Line 858: “insect NaV channels, comparing to the mammalian ones” instead “mammalian NaV channels, comparing to insect ones”.
Line 861: “Figure 8” instead “Figure 12”.
Line 862: “i.e.,” instead “i.e.”.
Line 866: “components of currents and slow down” instead “components of currents; they also slow down”.
Line 868: “Figure 8” instead “Figure 12”.
Line 867: Changes have been made to Figure 8.
Line 889: “residue, which results” instead “residue what results”.
Line 890: “of electrophysiological” instead “on electrophysiological”.
Line 895: “The sea anemone” instead “Sea anemone”.
Line 897: “NaV subtypes as well as” instead “NaV subtypes, as well as”.
Line 899: “of further design” instead “of a further design”.
Line 919: reference “Karabulut, A.; McClain, M.; Rubinstein, B.; Sabin, K.Z.; McKinney, S.A.; Gibson, M.C. The architecture and operating mechanism of a cnidarian stinging organelle. Nat. Commun. 2022, 13, 3494. doi: 10.1038/s41467-022-31090-0” instead “Fautin, D.F.; Mariscal, R.N. Cnidaria: Anthozoa. In Microscopic anatomy of invertebrates; Harrison, F.G., Westfall, J.A. Eds.; 919 Wiley-Liss, Inc: New York, 1991; Volume 2, pp. 267–358”.
Line 923: reference “Zenkert, C.; Takahashi, T.; Diesner, M.-O.; Özbek, S. Morphological and Molecular Analysis of the Nematostella vectensis Cnidom. PLoS ONE 2011, 6, e22725. doi: 10.1371/journal.pone.0022725.” instead “Fautin, D.G. Structural diversity, systematics, and evolution of cnidae. Toxicon 2009, 54, 1054–1064. doi: 923 10.1016/j.toxicon.2009.02.024.”
Lines 952, 971, 978, 1017, 1020, 1023, 1036, 1046, 1056, 1062, 1087, 1113, 1122, 1144, 1147, 1198, 1223, 1253, 1283, 1306, 1321, 1332, 1347: blue font color formed in the doi and/or surnames of the authors in the references of the original version of the manuscript [24, 32, 35, 50, 51, 52, 57, 61, 66, 69, 80, 90, 95, 104, 105, 121, 126, 137, 150, 163, 174, 180, 185, 192], we changed to black color.
Round 2
Reviewer 2 Report
The authors of this manuscript review the work on sodium channel modulating toxins over the past 40 years. This is an important review with the last extensive review on this topic occurring more than 10 years ago. The authors go into a lot of depth covering this topic in many aspects from sodium channels themselves to the evolution of genes encoind NaTx's too specific residues critical to the binding of the toxin to the channel.
While the work is mostly very nice, a lot of grammatical errors are still present. I have corrected some below but many still exist. In particular, the use of "ones" is out of place. The second major comment is the figures. I think the alignments are helpful but additional annotations of which are key residues would be helpful. Some sort of representation of the sodium channels would be also very useful and ideally how they interact with NaTx.
Line 9 : “ Many natural toxins and sea anemone ones among them (called neurotoxins) are an” Weird sentence and very long. What do you mean by sea anemone ones?
Line126-130 needs the reference for the date of 1980. Sentence is very long and confusing. Can you claifry which orgnamism you are talking about here (mammals or animals?)
Line 136 do you mean transmembrane domains. This sentence is confusing
Moran et al. (2009) staes that“α-subunit is ~260 kDa protein is composed of four homologous domains (D1–D4), each composed of comprising six transmembrane segments (S1–S6)”.
So, I think what is meant is that the α-subunit contains 4 domains comprised of six transmembrane segments (S1- S6). Segments S1–S4 form a voltage sensor and the two other segments (S5 and S6) participating in the formation of an ion-conducting pore (pore domain PD), permeable to sodium ions [37,38,59,61].
Line 143: “Currently, there are nine subtypes of human NaVs (NaV1.1–NaV1.9 [59,69]) encoded by the gene family (SCN1A–SCN5A and SCN8A–SCN11A)” while not totally wrong this is missing some important information. For example, a quote from de Lera Ruiz & Kraus (2015):
“In humans there are nine different α subunits, Nav1.1, Nav1.2, Nav1.3, Nav1.4, Nav1.5, Nav1.6, Nav1.7, Nav1.8, and Nav1.9, encoded by the genes SCN1A, SCN2A, SCN3A, SCN4A, SCN5A, SCN8A, SCN9A, SCN10A, and SCN11A, respectively. “
Therefore, the above sentence on line 143 please clarify that there are nine α subunits encoded by the gene family (SCN1A–SCN5A and SCN8A–SCN11A
Line 157 no end to brackets from “(a loop”
Line 210 what do you mean by equina mRNA ones
Line 325 drosophila should be capital and italicised
Line 356 not “received” maybe identified
Line 375 “testifies” seems like a strange word choice, maybe supports is better word
Reference 141 is from 2006, is this new?
This paragraph line 407 does not add much please remove or expand
This section was very nice and the authors built this into a nice story especially the overlap in function and of NaTx’s and other toxins like BDS toxins. Several studies have suggested that these 2 families might share a common evolutionary ancestor. I think expanding finishing this section by expanding this overlap and maybe combining it with evolutionary section would be beneficial
“3.2. Secondary and spatial structures of NaTxs” Bit strange to start a new section without some sort of intro, one sentence would be enough
Line 414: what do you mean by ββββ-type and this sentence lacks a reference
Figure 6 can you specify in the figure more clearly the β-defensin-like fold and the Arg14 loop’ consisting of 7–18 aa.
Line 449: “In the late 1960s, it was” no references are from 1960’s?
Line 469 bracket doesn’t end
Line 475 “At the same time, almost all representatives of these NaTxs are characterized by the presence of a unique N-terminal 4-membered fragment, 45RKKK48.” I think you mean C- terminus
The description of key residues in the paragraph from line 466 is hard to follow. Can the authors maybe highlight these residues on some example sequences and 3d structures. I find myself having to go back to figure 1 many times and it’s hard to keep track. A figure with the Nav as well highlighting its features would also be beneficial. And how some residues are key for this interaction.
Line 597 can you expand on this synergy
Line 602 what study
Line 612 significant ones?
Line 659 “their lethal doses for crabs being practically in the same range of values”
What do you mean practically in the same range of value. Maybe give the range you mean because they do vary quite a lot but less than mammals for sure.
Line 711 reference
Line 757 figure11 ?
Can you annotate the figure 7 neurotoxin more? Maybe mark were the 2 chains start and stop
For Table 3 why were these NaTx’s included? Only δ-SHTX-Hcr1f, RTX-III and RTX-VI have tested against BgNaV1. It makes it look like no NaTx’s have tested against this channel before. But this is not the case. For example, Nv1, Nv4 and Nv5 have all been tested against BgNaV1(Sachkova et al., 2019). APETx3 is known for interacting with Nav1.2–1.4, Nav1.6, DmNav1, BgNav1, and VdNav1 (Peigneur et al., 2012). Maybe expand this list in this table.
Where are the EC values for BgNav1, and VdNav1 in table 4. Because of this I don’t understand the comment line 825
Author Response
The authors of this manuscript review the work on sodium channel modulating toxins over the past 40 years. This is aЕn important review with the last extensive review on this topic occurring more than 10 years ago. The authors go into a lot of depth covering this topic in many aspects from sodium channels themselves to the evolution of genes encodind NaTx's too specific residues critical to the binding of the toxin to the channel.
While the work is mostly very nice, a lot of grammatical errors are still present. I have corrected some below but many still exist. In particular, the use of "ones" is out of place. The second major comment is the figures. I think the alignments are helpful but additional annotations of which are key residues would be helpful. Some sort of representation of the sodium channels would be also very useful and ideally how they interact with NaTx.
In particular, the use of "ones" is out of place.
Answer:
We have removed this word from the text on lines 95, 115, 210, 464, 554, 612, 671, 787, 871 and replaced it with words corresponding in meaning.
The second major comment is the figures. I think the alignments are helpful but additional annotations of which are key residues would be helpful. Some sort of representation of the sodium channels would be also very useful and ideally how they interact with NaTx.
Answer:
This interaction of one of the representatives of Heteractis neurotoxins, δ-SHTX-Hcr1f, with the Nav1.3 model [168] with a description of key functionally significant residues, we briefly described in chapter 5 of this manuscript.
Line 9: “Many natural toxins and sea anemone ones among them (called neurotoxins) are an” Weird sentence and very long. What do you mean by sea anemone ones?
Answer:
Line 9: The long sentence “Many natural toxins and sea anemone ones among them (called neurotoxins) are an indispensable and promising tool in pharmacological researches, which have widely been carried out over the past three decades, in particular, in establishing different NaV subtypes functional properties and a specific role in various pathologies“ was changed on two sentence:
“Many natural toxins including the sea anemone toxins (called neurotoxins) are an indispensable and promising tool in pharmacological researches. They have widely been carried out over the past three decades, in particular, in establishing different NaV subtypes functional properties and a specific role in various pathologies.”
and the word “ones“ was changed on word “toxins”.
Line126-130 needs the reference for the date of 1980. Line126-130: Sentence is very long and confusing. Can you clarify which organism you are talking about here (mammals or animals?)
Answer:
The relevant references [21,23-30] have been inserted.
We have divided the first sentence into two.
The spatial structure of described NaV is typical for all eukaryotes (both mammals and animals). In the review cited (reference [63]: Marban et al. Structure and function of voltage-gated sodium channels. J. Physiol. 1998, 508, 1065 647–657), the authors referred to the first structural study of only the axonal NaV.
Moran et al. (2009) staes that“α-subunit is ~260 kDa protein is composed of four homologous domains (D1–D4), each composed of comprising six transmembrane segments (S1–S6)”.
So, I think what is meant is that the α-subunit contains 4 domains comprised of six transmembrane segments (S1-S6). Segments S1–S4 form a voltage sensor and the two other segments (S5 and S6) participating in the formation of an ion-conducting pore (pore domain PD), permeable to sodium ions [37,38,59,61]. //
Answer:
Yes, of course. You are right; we accompany this information in this paragraph with reference on Moran et al. (2009) [37].
Line 143: “Currently, there are nine subtypes of human NaVs (NaV1.1–NaV1.9 [59,69]) encoded by the gene family (SCN1A–SCN5A and SCN8A–SCN11A)” while not totally wrong this is missing some important information. For example, a quote from de Lera Ruiz & Kraus (2015):
“In humans there are nine different α subunits, Nav1.1, Nav1.2, Nav1.3, Nav1.4, Nav1.5, Nav1.6, Nav1.7, Nav1.8, and Nav1.9, encoded by the genes SCN1A, SCN2A, SCN3A, SCN4A, SCN5A, SCN8A, SCN9A, SCN10A, and SCN11A, respectively.“
Therefore, the above sentence on line 143 please clarify that there are nine α subunits encoded by the gene family (SCN1A–SCN5A and SCN8A–SCN11A.
Answer:
Line 143: Dear reviewer, thank you very much for the reference with an excellent publication by these authors. We corrected the paragraph a little and replaced reference [38] (Cardoso, F.C.; Lewis, R.J. Sodium channels and pain: From toxins to therapies. Br. J. Pharmacol. 2018, 175, 2138–2157. doi: 984 10.1111/bph.13962.) with reference (de Lera Ruiz, M.; Kraus, R.L. Voltage-Gated Sodium Channels: Structure, Function, Pharmacology, and Clinical Indications. J. Med. Chem. 2015, 58, 7093–7118.).
Line 210 what do you mean by equina mRNA ones?
Answer:
the word “ones” is replaced by the word “sequences”.
Line 325 drosophila should be capital and italicized.
Answer:
Dear reviewer, we did not understand this remark, since the name of Drosophila mentioned in the text was not given.
Line 334: The name of the sea anemone Anemonia erythraea is in italics.
Line 334: The name Phoneutria nigriventer is in italics.
Line 356 not “received” maybe identified
Answer:
The word “received” was changed on “identified”.
Line 375 “testifies” seems like a strange word choice, maybe supports is better word
Answer: the word “testifies” was changed on the word “supports”.
Reference 141 is from 2006, is this new?
Answer:
The word “new” from 2006 reference [141] (Oliveira et al. 2006) refers not to the year, but to the new BcIV peptide (relative to BcIII type 1 neurotoxin from Bunodosoma caissarum [104]), which was identified in this sea anemone 13 years later. BcIV cannot be attributed to the type 1 neurotoxins (lower molecular weight, homology and toxicity); it can be attributed to the sea anemone APETx-like peptide. Nevertheless, the paralyzing effect of BcIV makes it possible to draw similarities between BcIV and BcIII in terms of toxicity. Thus, the word "new" is, in our opinion, quite appropriate for BcIV.
Line 407: This paragraph line 407 does not add much please remove or expand
Answer:
We agree with the respected reviewer and have removed this sentence from the text: “ Recently two unique sequences (the toxins Avi 9a-1 and Avi 9a-2) have been found in the transcriptome of the sea anemone A. viridis [88]. They are four- and three-repeated structures homologous to the toxin Am-I (27 aa) from the sea anemone A. maculata [97].”
This section was very nice and the authors built this into a nice story especially the overlap in function and of NaTx’s and other toxins like BDS toxins. Several studies have suggested that these 2 families might share a common evolutionary ancestor. I think expanding finishing this section by expanding this overlap and maybe combining it with evolutionary section would be beneficial
Answer:
We thank the distinguished reviewer for this suggestion, we will try to fulfill it, but in our next work. Since we need to establish the genome sequences, phylogenetic and structure-functional relationships of currently known BDS toxins, APETx-like peptides and the neurotoxins described in this review. We are currently conducting electrophysiological studies of several native APETx-like peptides from H. crispa, as well as their recombinant analogues and their pharmacological potential.
Yes, indeed, despite the common β-defensin-like fold of neurotoxins and APETx-like peptides, their biological targets are acid-sensitive ASIC channels [41-44] and the acetylcholine receptor [Kalina et al. Nicotinic Acetylcholine Receptors Are Novel Targets of APETx-like Toxins from the Sea Anemone Heteractis magnifica. Toxins 2022, 14(10), 697; https://doi.org/10.3390/toxins14100697], as we reported in a recent paper. Therefore, now, our efforts are aimed at elucidation of targets for H. crispa APETx-like peptide, which will be useful and necessary for their phylogenetic analysis and identification of an evolutionary ancestor.
“3.2. Secondary and spatial structures of NaTxs” Bit strange to start a new section without some sort of intro, one sentence would be enough
Answer:
Line 411: at the beginning of the first paragraph, we added the sentence “The search and study of NaTx biological targets and the elucidation of their structure-functional relationships are impossible without determination of the spatial structure of neurotoxins of all structural groups. The 3D structure, in turn, is the basis for studying the mechanisms of protein-ligand interactions.”
Line 414: what do you mean byVs lj,fdbkb ββββ-type and this sentence lacks a reference
Answer:
Line 414:
We inserted reference [118] at the end of this sentence and changed the phrase "ββββ-type" to " β-defensin-like type".
Figure 6 can you specify in the figure more clearly the β-defensin-like fold and the Arg14 loop’ consisting of 7–18 aa.
Answer:
We made the changes proposed by the reviewer to the structures of neurotoxins in this figure, who became number 7.
Line 449: “In the late 1960s, it was” no references are from 1960’s?
Answer:
Line 449:
We changed the phrase "In the late 60s " to "In the late 1960-80s".
The references corresponding to years are at the end of the sentence.
Line 464:
We changed the word "ones" to "residues".
Line 469 bracket doesn’t end
Answer:
We closed the parenthesis after (Figure 1)).
Line 475 “At the same time, almost all representatives of these NaTxs are characterized by the presence of a unique N-terminal 4-membered fragment, 45RKKK48.” I think you mean C- terminus
Answer:
Yes, you are right. We changed the “N-terminal 4-membered fragment” to “C-terminal 4-membered fragment”.
The description of key residues in the paragraph from line 466 is hard to follow. Can the authors maybe highlight these residues on some example sequences and 3d structures. I find myself having to go back to figure 1 many times and it’s hard to keep track. A figure with the Nav as well highlighting its features would also be beneficial. And how some residues are key for this interaction.
Answer:
Lines 147 and 532: We have added (at the request of a respected reviewer) 3D-models of the structure of the Nav channel (Figure 1), as well as neurotoxins with some stained functionally significant residues (Figure 8). We changed the numbering of all figures, captions and numbering of references to figures according to the text.
As for the functionally significant residues that are key to the interaction of NaTx with Navs, some of them are mentioned in chapter “3.3. Some key functionally significant amino acid residues of NaTxs”, as well as chapter “3.4. Toxicity of NaTxs” and chapter “5. Interaction of NaTxs with NaV subtypes tested electrophysiologically and by molecular docking”. It is obvious that key functionally significant amino acid residues differ in different representatives of neurotoxins, as they have undergone evolutionary diversification aimed at interacting with various biological targets, including different Navs subtypes.
Line 581:
We changed (Fig. 6) to (Figure 6).
Line 597 can you expand on this synergy
Answer:
Obviously, neurotoxins of different structural types act synergistically on target organisms. As noted by the authors of the publication [37] (Moran et al.): “The combined of these compounds (M.M.: neurotoxins of different types of NaTxs) has apparently been successful over hundreds of millions of years as is evident by the ability of sea anemones to colonize and thrive in a wide variety of ecological niches. Moreover, the ever changing environment and appearance of new species has probably enforced diversification of toxins in sea anemones”.
Line 602 what study
Answer:
We inserted the phrase “the results of the”mechanical and electrophysiological study” instead of the phrase “the results of the study”.
Line 612 significant ones?
Answer:
We inserted the phrase “significant residues” inserted the phrase “significant ones“.
Line 659 “their lethal doses for crabs being practically in the same range of values”.
What do you mean practically in the same range of value. Maybe give the range you mean because they do vary quite a lot but less than mammals for sure.
Answer:
instead of the phrase "their lethal doses for crabs are practically in the same range of values" we inserted the phrase "their lethal doses for crabs being in the range from 0.6 (ShI) to 90.0 (RpIV) of values”.
Line 711 reference
Answer:
The sentence "The spatial structure of 1H NMR RTX-III determined by Hinds and Norton [183] was similar to that of RpII [27]" we replaced on the sentence "The spatial structure of 1H NMR RTX-III determined by Hinds and Norton [183] was similar to the 1H NMR structure of another type 2 neurotoxin, RpII [27].”
Line 757 figure 11 ?
Answer:
Figure 11 replaced with Figure 7.
Can you annotate the figure 7 neurotoxin more? Maybe mark were the 2 chains start and stop
Answer:
Line 766: we removed the hyphen that was mistakenly placed in the alignment to the position of the missing Arg13.
Therefore, in the caption to Figure 8, we added the phrase “Amino acid sequence of RTX-VI includes two peptides (residues 1-12 and 14-48), both peptides are connected by two disulfide bridges, C1-C5, C2-C4.”
For Table 3 why were these NaTx’s included? Only δ-SHTX-Hcr1f, RTX-III and RTX-VI have tested against BgNaV1. It makes it look like no NaTx’s have tested against this channel before. But this is not the case. For example, Nv1, Nv4 and Nv5 have all been tested against BgNaV1(Sachkova et al., 2019). APETx3 is known for interacting with Nav1.2–1.4, Nav1.6, DmNav1, BgNav1, and VdNav1 (Peigneur et al., 2012). Maybe expand this list in this table.
Line 793
Answer:
We have included several additional neurotoxins in Table 3. Their names are included in the table of contents.
Where are the EC values for BgNav1, and VdNav1 in table 4. Because of this I don’t understand the comment line 825
Line 821:
Answer:
We have added some EC values for BgNav1 and VdNav1 to Table 4.